# Improved Object-Centric Diffusion Learning with Registers and Contrastive Alignment

**Bac Nguyen**[1][*]**, Yuhta Takida**[1]**, Naoki Murata**[1]**, Chieh-Hsin Lai**[1]**, Toshimitsu Uesaka**[1]**,
Stefano Ermon**[2]**, Yuki Mitsufuji**[1,3]

[1]Sony AI, [2]Stanford University, [3]Sony Group Corporation

## Abstract

Slot Attention (SA) with pretrained diffusion models has recently shown promise for object-centric learning (OCL), but suffers from *slot entanglement* and *weak alignment* between object slots and image content. We propose **C**ontrastive **O**bject-centric **D**iffusion **A**lignment (`CODA`), a simple extension that (i) employs register slots to absorb residual attention and reduce interference between object slots, and (ii) applies a contrastive alignment loss to explicitly encourage slot–image correspondence. The resulting training objective serves as a tractable surrogate for maximizing mutual information (MI) between slots and inputs, strengthening slot representation quality. On both synthetic (MOVi-C/E) and real-world datasets (VOC, COCO), `CODA` improves object discovery (e.g., +6.1% FG-ARI on COCO), property prediction, and compositional image generation over strong baselines. Register slots add negligible overhead, keeping `CODA` efficient and scalable. These results indicate potential applications of `CODA` as an effective framework for robust OCL in complex, real-world scenes. Code and pretrained models are available at https://github.com/sony/coda.

## 1 Introduction

Object-centric learning (OCL) aims to decompose complex scenes into structured, interpretable object representations, enabling downstream tasks such as visual reasoning (Assouel et al., 2022; D'Amario et al., 2021), causal inference (Schölkopf et al., 2021; Zholus et al., 2022), world modeling (Ke et al., 2021), robotic control (Haramati et al., 2024), and compositional generation (Singh et al., 2022a). Yet, learning such compositional representations directly from images remains a core challenge. Unlike text, where words naturally form composable units, images lack explicit boundaries for objects and concepts. For example, in a street scene with pedestrians, cars, and traffic lights, a model must disentangle these entities without labels and also capture their spatial relations (e.g., a person crossing in front of a car). Multi-object scenes add further complexity: models must not only detect individual objects but also capture their interactions. As datasets grow more cluttered and textured, this becomes even harder. Manual annotation of object boundaries or compositional structures is costly, motivating the need for fully unsupervised approaches such as Slot Attention (SA) (Locatello et al., 2020). While effective in simple synthetic settings, SA struggles with large variations in real-world images, limiting its applicability to visual tasks such as image or video editing.

Combining SA with diffusion models has recently pushed forward progress in OCL (Jiang et al., 2023; Wu et al., 2023; Akan & Yemez, 2025). In particular, Stable-LSD (Jiang et al., 2023) and SlotAdapt (Akan & Yemez, 2025) achieve strong object discovery and high-quality generation by leveraging pretrained diffusion backbones such as Stable Diffusion (Rombach et al., 2022) (SD). Nevertheless, these approaches still face two key challenges. First, as illustrated in Fig. 1 (left), they often suffer from **slot entanglement**, where a slot encodes features from multiple objects or fragments of them, leading to unfaithful single-slot generations. This entanglement degrades segmentation quality and prevents composable generation to novel scenes and object configurations. Second, they exhibit **weak alignment**, where slots fail to consistently correspond to distinct image regions, especially on real-world images. As shown in our experiments, slots often suffer from over-segmentation (splitting one object into multiple slots), under-segmentation (merging multiple

---

[*]Correspondence to bac.nguyencong@sony.com

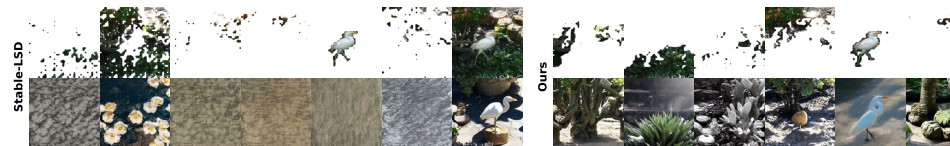

Figure 1: Image generation from individual slots. **Top:** slot masks. **Bottom:** generated images. Both methods can reconstruct the full scene when conditioned on all slots (last column). However, Stable-LSD (without register slots) fails to generate images from individual slots. Our method yields faithful single-concept generations, demonstrating disentangled and well-aligned slots.

objects into one slot), or inaccurate object boundaries. Together, these two issues reduce both the accuracy of object-centric representations and their utility for compositional scene generation.

In response, we propose **C**ontrastive **O**bject-centric **D**iffusion **A**lignment (`CODA`), a slot-attention model that uses a pretrained diffusion decoder to reconstruct the input image. `CODA` augments the model with register slots, which absorb residual attention and reduce interference between object slots, and a contrastive objective, which explicitly encourages slot–image alignment. As illustrated in Fig. 1 (right), `CODA` faithfully generates images from both individual slots as well as their compositions. In summary, the contributions of this paper can be outlined as follows.

(i) **Register-augmented slot diffusion.** We employ register slots that are independent of the input image into slot diffusion. Although these register slots carry no semantic information, they act as attention sinks, absorbing residual attention mass so that semantic slots remain focused on meaningful object–concept associations. This reduces interference between object slots and mitigating slot entanglement (Section 4.1).

(ii) **Mitigating text-conditioning bias.** To reduce the influence of text-conditioning biases inherited from pretrained diffusion models, we finetune the key, value, and output projections in cross-attention layers. This adaptation further improves alignment between slots and visual content, ensuring more faithful object-centric decomposition (Section 4.2).

(iii) **Contrastive alignment objective.** We propose a contrastive loss that ensures slots capture concepts present in the image (Section 4.3). Together with the denoising loss, our training objective can be viewed as a tractable surrogate for maximizing the mutual information (MI) between inputs and slots, improving slot representation quality (Section 4.4).

(iv) **Comprehensive evaluation.** We demonstrate that `CODA` outperforms existing unsupervised diffusion-based approaches across synthetic and real-world benchmarks in object discovery (Section 5.1), property prediction (Section 5.2), and compositional generation (Section 5.3). On the VOC dataset, `CODA` improves instance-level object discovery by +3.88% $mBO^i$ and +3.97% $mIoU^i$, and semantic-level object discovery by +5.72% $mBO^c$ and +7.00% $mIoU^c$. On the COCO dataset, it improves the foreground Adjusted Rand Index (FG-ARI) by +6.14%.

## 2 RELATED WORK

**Object-centric learning (OCL).** The goal of OCL is to discover compositional object representations from images, enabling systematic generalization and stronger visual reasoning (D'Amario et al., 2021; Assouel et al., 2022). Learning directly from raw pixels is difficult, so previous works leveraged weak supervision (e.g., optical flow (Kipf et al., 2022), depth (Elsayed et al., 2022), text (Xu et al., 2022), pretrained features (Seitzer et al., 2023)), or auxiliary losses that guide slot masks toward moving objects (Bao et al., 2022; 2023; Zadaianchuk et al., 2023). Scaling OCL to complex datasets has been another focus: DINOSAUR (Seitzer et al., 2023) reconstructed self-supervised features to segment real-world images, and FT-DINOSAUR (Didolkar et al., 2025) extended this via encoder finetuning for strong zero-shot transfer. SLATE (Singh et al., 2022a) and STEVE (Singh et al., 2022b) combined discrete VAE tokenization with slot-conditioned autoregressive transformers, while SPOT (Kakogeorgiou et al., 2024) improved autoregressive decoders using patch permutation and attention-based self-training. Our work builds on SA, but does not require any additional supervision.

**Diffusion models for OCL.** Recent works explored diffusion models (Sohl-Dickstein et al., 2015; Rombach et al., 2022) as slot decoders in OCL. Different methods vary in how diffusion models

are integrated. For example, SlotDiffusion (Wu et al., 2023) trained a diffusion model from scratch, while Stable-LSD (Jiang et al., 2023), GLASS (Singh et al., 2025), and SlotAdapt (Akan & Yemez, 2025) leveraged pretrained diffusion models. Although pretrained models offer strong generative capabilities, they are often biased toward text-conditioning. To address this issue, GLASS (Singh et al., 2025) employed cross-attention masks as pseudo-ground truth to guide SA training. Unlike GLASS, `CODA` does not rely on supervised signals such as generated captions. SlotAdapt (Akan & Yemez, 2025) introduced adapter layers to enable new conditional signals while keeping the base diffusion model frozen. In contrast, `CODA` simply finetunes key, value, and output projections in cross-attention, without introducing additional layers. This ensures full compatibility with off-the-shelf diffusion models while remaining conceptually simple and computationally efficient.

**Contrastive learning for OCL.** Training SA with only reconstruction losses often leads to unstable or inconsistent results (Kim et al., 2023). To improve robustness, several works introduced contrastive objectives. For example, SlotCon (Wen et al., 2022) applied the InfoNCE loss (Oord et al., 2018) across augmented views of the input image to enforce slot consistency. Manasyan et al. (2025) used contrastive loss to enforce the temporal consistency for video object-centric models. In contrast, `CODA` tackles compositionality by aligning images with their slot representations, enabling faithful generation from both individual slots and their combinations. Unlike Jung et al. (2024), who explicitly maximize likelihood under random slot mixtures and thus directly tune for compositional generation, `CODA` focuses on enforcing slot–image alignment; its gains in compositionality arise indirectly from improved disentanglement. Although `CODA` uses a negative loss term, similar to negative guidance in diffusion models (Karras et al., 2024), the roles are fundamentally different. Karras et al. (2024) apply negative guidance during sampling to steer the denoising trajectory, whereas `CODA` uses a contrastive loss during training to improve slot–image alignment.

## 3 BACKGROUND

**Slot Attention** (Locatello et al., 2020) **(SA).** Given input features $\mathbf{f} \in \mathbb{R}^{M \times D_{\text{input}}}$ of an image, the goal of OCL is to extract a sequence $\mathbf{s} \in \mathbb{R}^{N \times D_{\text{slot}}}$ of $N$ slots, where each slot is a $D_{\text{slot}}$-dimensional vector representing a composable concept. In SA, we start with randomly initialized slots as $\mathbf{s}^{(0)} \in \mathbb{R}^{N \times D_{\text{slot}}}$. Once initialized, SA employs an iterative mechanism to refine the slots. In particular, slots serve as *queries*, while the input features serve as *keys* and *values*. Let $q$, $k$, and $v$ denote the respective linear projections used in the attention computation. Given the current slots $\mathbf{s}^{(t)}$ and input features $\mathbf{f}$, the update rule can be formally described as

$$\mathbf{s}^{(t+1)} = \texttt{GRU}(\mathbf{s}^{(t)}, \mathbf{u}^{(t)}) \quad \text{where} \quad \mathbf{u}^{(t)} = \texttt{Attention}(q(\mathbf{s}^{(t)}), k(\mathbf{f}), v(\mathbf{f})).$$

Here, attention readouts are aggregated and refined through a Gated Recurrent Unit (Cho et al., 2014) (GRU). Unlike self-attention (Vaswani et al., 2017), the `softmax` function in SA is applied along the slot axis, enforcing competition among slots. This iterative process is repeated for several steps, and the slots from the final iteration are taken as the slot representations. Finally, these slots are passed to a decoder trained to reconstruct the input image. The slot decoder can take various forms, such as an MLP (Watters et al., 2019) or an autoregressive Transformer (Vaswani et al., 2017). Interestingly, recent works (Jiang et al., 2023; Singh et al., 2025; Akan & Yemez, 2025) have shown that using (latent) diffusion models as slot decoders proves to be particularly powerful and effective in OCL.

**Latent diffusion models** (Rombach et al., 2022) **(LDMs).** Diffusion models are probabilistic models that sample data by gradually denoising Gaussian noise (Sohl-Dickstein et al., 2015; Song et al., 2021; Ho et al., 2020). The forward process progressively corrupts data with Gaussian noise, while the reverse process learns to denoise and recover the original signal. To improve efficiency, SD performs this process in a compressed latent space rather than pixel space. Concretely, a pretrained autoencoder maps an image to a latent vector $\mathbf{z} \in \mathcal{Z}$, where a U-Net denoiser iteratively refines noisy latents. Consider a variance preserving process that mixes the signal $\mathbf{z}$ with Gaussian noise $\boldsymbol{\epsilon} \sim \mathcal{N}(0, \mathbf{I})$, given by $\mathbf{z}_\gamma = \sqrt{\sigma(\gamma)}\mathbf{z} + \sqrt{\sigma(-\gamma)}\boldsymbol{\epsilon}$, where $\sigma(.)$ is the sigmoid function and $\gamma$ is the log signal-to-noise ratio. Let $\boldsymbol{\epsilon_\theta}(\mathbf{z}_\gamma, \gamma, \mathbf{c})$ denote a denoiser parameterized by $\boldsymbol{\theta}$ that predicts the Gaussian noise $\boldsymbol{\epsilon}$ from noisy latents $\mathbf{z}_\gamma$, conditioned on an external signal $\mathbf{c}$. In SD, conditioning is implemented through cross-attention, which computes attention between the conditioning signal and the features produced by U-Net. Training diffusion models is formulated as a noise prediction

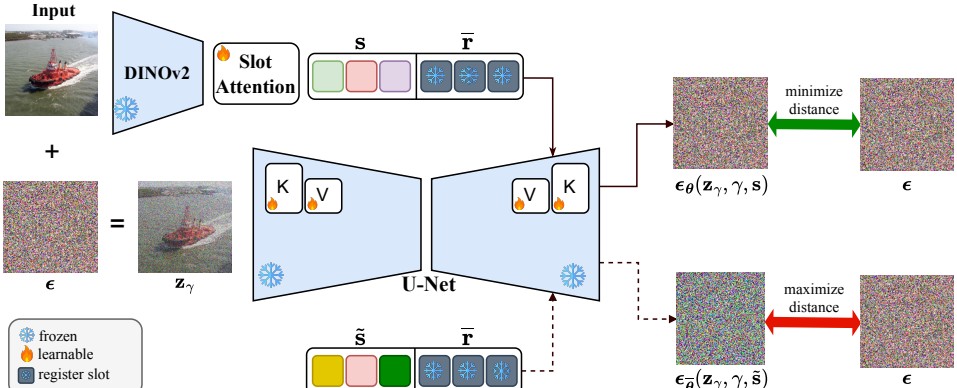

Figure 2: **Overview of CODA.** The input image is encoded with DINOv2 and processed by Slot Attention (SA) to produce slot representations. The semantic slots $\mathbf{s}$, together with register slots $\overline{\mathbf{r}}$, serve as conditioning inputs for the cross-attention layers of a pretrained diffusion model. SA is trained jointly with the key, value, and output projections of the cross-attention layers using a denoising objective that minimizes the mean squared error between the true and predicted noise. In addition, a contrastive loss is applied to align each image with its corresponding slot representations.

problem, where the model learns to approximate the true noise $\boldsymbol{\epsilon}$ added during the forward process,

$$\min_{\boldsymbol{\theta}} \quad \mathbb{E}_{(\mathbf{z},\mathbf{c}),\boldsymbol{\epsilon},\gamma} \left[ \| \boldsymbol{\epsilon} - \boldsymbol{\epsilon}_{\boldsymbol{\theta}}(\mathbf{z}_\gamma, \gamma, \mathbf{c}) \|_2^2 \right]$$

with $(\mathbf{z}, \mathbf{c})$ sampled from a data distribution $p(\mathbf{z}, \mathbf{c})$. Once training is complete, sampling begins from random Gaussian noise, which is iteratively refined using the trained denoiser.

## 4 PROPOSED METHOD

As summarized in Fig. 2, CODA builds on diffusion-based OCL by extracting slot sequences from DI-NOv2 features (Oquab et al., 2024) with SA and decoding them using a pretrained SD v1.5 (Rombach et al., 2022). To address the challenges of slot entanglement and weak alignment, CODA introduces three components: (i) register slots to absorb residual attention and keep object slots disentangled, (ii) finetuning of cross-attention keys and queries to mitigate text-conditioning bias, and (iii) a contrastive alignment loss to explicitly align images with their slots. These modifications yield disentangled, well-aligned slots that enable faithful single-slot generation and compositional editing.

### 4.1 REGISTER SLOTS

An ideal OCL model should generate semantically faithful images when conditioned on arbitrary subsets of slots. In practice, however, most diffusion-based OCL methods fall short of this goal. As discussed in the introduction, decoding a single slot typically yields distorted or semantically uninformative outputs. Although reconstructions from the full set of slots resemble the input images, this reliance reveals a strong interdependence among slots (see Section D for more detailed analysis). Such slot entanglement poses a challenge for compositionality, particularly when attempting to reuse individual concepts in novel configurations.

To address this problem, we add input-independent register slots that act as residual attention sinks, absorbing shared or background information and preventing object slots from mixing. Intuitively, register slots are semantically empty but structurally valid inputs, making them natural placeholders for slots that can capture residual information without competing with object representations. We obtain these register slots by passing only padding tokens through the SD text encoder, a pretrained ViT-L/14 CLIP (Radford et al., 2021). Formally, let pad denote the padding token used to ensure fixed-length prompts in text-to-image SD. By encoding the sequence [pad, ..., pad] with the frozen text encoder[1], we obtain a fixed-length sequence of frozen embeddings serving as register slots $\overline{\mathbf{r}}$.

---

[1]For SD v1.5, 77 padding tokens are used, resulting in 77 register slots.

We also explore an alternative design with trainable register slots in Section E.2, and find that while learnable registers can also mitigate slot entanglement, our simple fixed registers perform best.

**Why do register slots mitigate slot entanglement?** The `softmax` operation in cross-attention forces attention weights to sum to one across all slots. When a query from U-Net features does not strongly match any semantic slot, this constraint causes the attention mass to spread arbitrarily, weakening slot–concept associations. Register slots serve as placeholders that absorb this residual attention, giving the model extra capacity to store auxiliary information without interfering with semantic slots. This leads to cleaner and more coherent slot-to-concept associations. Consistent with this view, we observe in Section C that a substantial fraction of attention mass is allocated to register slots. A similar phenomenon has been reported in language models, where `softmax` normalization causes certain initial tokens to act as attention sinks (Xiao et al., 2024; Gu et al., 2025), absorbing unnecessary attention mass and preventing it from distorting meaningful associations.

In a related approach, Akan & Yemez (2025) introduced an additional embedding by pooling from either generated slots or image features. Unlike our method, their embedding is injected directly into the cross-attention layers and is explicitly designed to capture global scene information. While this might provide contextual guidance, it ties the model to input-specific features, reducing flexibility in reusing slots across arbitrary compositions. In contrast, our register slots are independent of the input image, making them better suited for compositional generation.

## 4.2 FINETUNING CROSS-ATTENTION

SD is trained on large-scale image–text pairs, so directly using its pretrained model as a slot decoder introduces a text-conditioning bias: the model expects text embeddings and tends to prioritize language-driven semantics over slot-level representations (Akan & Yemez, 2025). This mismatch weakens the fidelity of slot-based generation. Prior works have approached this issue in different ways. For example, Wu et al. (2023) trained diffusion models from scratch, thereby removing text bias but sacrificing generative quality due to limited training data. More recently, Akan & Yemez (2025) proposed adapter layers (Mou et al., 2024) to align slot representations with pretrained diffusion models, retaining generation quality but still relying on text-conditioning features.

In contrast, we adopt a lightweight adaptation strategy: finetuning only the key, value, and output projections in cross-attention layers (Kumari et al., 2023). This allows the model to better align slots with visual content, mitigating text-conditioning bias while preserving the expressive power of the pretrained diffusion backbone. We find this minimal modification sufficient to eliminate the bias introduced by text conditioning. Unlike the previous approaches, our method is both computationally and memory efficient, requiring no additional layers or architectural modifications. This makes our approach not only effective but also conceptually simple. Formally, let $\phi$ denote the parameters of SA, the denoising objective for diffusion models can be written as

$$\mathcal{L}_{\mathrm{dm}}(\boldsymbol{\phi}, \boldsymbol{\theta}) = \mathbb{E}_{(\mathbf{z},\mathbf{s}),\boldsymbol{\epsilon},\gamma} \left[ \|\boldsymbol{\epsilon} - \boldsymbol{\epsilon}_{\boldsymbol{\theta}}(\mathbf{z}_{\gamma}, \gamma, \mathbf{s}, \bar{\mathbf{r}})\|_2^2 \right] , \qquad (1)$$

where $(\mathbf{z}, \mathbf{s})$ are sampled from $p(\mathbf{z})p_{\phi}(\mathbf{s} \mid \mathbf{z})$. In practice, $\mathbf{s}$ is not computed directly from $\mathbf{z}$, but rather from DINOv2 features of the image corresponding to $\mathbf{z}$. The U-Net is conditioned on the concatenation $(\mathbf{s}, \bar{\mathbf{r}})$ of semantic slots $\mathbf{s}$ and register slots $\bar{\mathbf{r}}$. During training, the parameters of SA are optimized jointly with the finetuned key, value, and output projections of SD, while other parameters are kept frozen.

## 4.3 CONTRASTIVE ALIGNMENT

The goal of OCL is to learn composable slots that capture distinct concepts from an image. However, in diffusion-based OCL frameworks, slot conditioning only serves as auxiliary information for the denoising loss, providing no explicit supervision to ensure that slots capture concepts present in the image. As a result, slots may drift toward arbitrary or redundant representations, limiting their interpretability and compositionality.

To address this, we propose a contrastive alignment objective that explicitly aligns slots with image content while discouraging overlap between different slots. Intuitively, the model should assign high likelihood to the correct slot representations and low likelihood to mismatched (negative) slots. Concretely, in addition to the standard denoising loss in Eq. (1), we introduce a contrastive loss

defined as the negative of denoising loss evaluated with negative slots $\tilde{\mathbf{s}}$:

$$\mathcal{L}_{\text{cl}}(\boldsymbol{\phi}) = -\mathbb{E}_{(\mathbf{z}, \tilde{\mathbf{s}}), \boldsymbol{\epsilon}, \gamma} \left[ \| \boldsymbol{\epsilon} - \boldsymbol{\epsilon}_{\overline{\boldsymbol{\theta}}}(\mathbf{z}_\gamma, \gamma, \tilde{\mathbf{s}}, \overline{\mathbf{r}}) \|_2^2 \right] , \tag{2}$$

where $(\mathbf{z}, \tilde{\mathbf{s}})$ are sampled from $p(\mathbf{z})q_\phi(\tilde{\mathbf{s}} \mid \mathbf{z})$ and $\overline{\boldsymbol{\theta}}$ denotes stop-gradient parameters of $\boldsymbol{\theta}$. Minimizing Eq. (1) increases likelihood under aligned slots, while minimizing Eq. (2) decreases likelihood under mismatched slots. We freeze the diffusion decoder and update only the SA module in Eq. (2), preventing the decoder from trivially reducing contrastive loss by altering its generation process. This ensures that improvements come from better slot representations rather than shortcut solutions. As confirmed by our ablations (see Table 5), unfreezing the decoder leads to unstable training and degraded performance across all metrics.

Finally, combining Eqs. (1) and (2), the overall training objective of CODA is defined as

$$\mathcal{L}(\boldsymbol{\phi}, \boldsymbol{\theta}) = \mathcal{L}_{\text{dm}}(\boldsymbol{\phi}, \boldsymbol{\theta}) + \lambda_{\text{cl}} \mathcal{L}_{\text{cl}}(\boldsymbol{\phi}) , \tag{3}$$

where $\lambda_{\text{cl}} \geq 0$ controls the trade-off between the denoising and contrastive terms. We study the effect of varying $\lambda_{\text{cl}}$ in Section E.4. This joint objective forms a contrastive learning scheme that acts as a surrogate for maximizing the MI between slots and images, as further discussed in Section 4.4.

**Strategy for composing negative slots.** A straightforward approach for obtaining negative slots is to sample them from unrelated images. However, such negatives are often too trivial for the decoder, providing little useful gradient signal. To address this, we construct *hard negatives*—more informative mismatches that push the model to refine its representations more effectively (Robinson et al., 2021). Concretely, given two slot sequences, $\mathbf{s}$ and $\mathbf{s}'$, extracted from distinct images $\mathbf{x}$ and $\mathbf{x}'$, we form negatives for $\mathbf{x}$ by randomly replacing a subset of slots in $\mathbf{s}$ with slots from $\mathbf{s}'$. This produces mixed slot sets that only partially match the original image, creating harder and more instructive negative examples. In our experiments, we replace half of the slots in $\mathbf{s}$ with those from $\mathbf{s}'$, and provide an ablation over different replacement ratios in Section E.5. A remaining challenge is that naive mixing can yield invalid combinations, e.g., omitting background slots or combining objects with incompatible shapes or semantics. To mitigate this, we share the slot initialization between $\mathbf{x}$ and $\mathbf{x}'$. Because initialization is correlated with the objects each slot attends to, sampling from mutually exclusive slots under shared initialization is more likely to produce semantically valid negatives than purely random mixing (Jung et al., 2024).

## 4.4 CONNECTION WITH MUTUAL INFORMATION

A central goal of our framework is to maximize MI between slots and the input image, so that slots capture representations that are both informative and compositional. To make this connection explicit, we reinterpret our training objective in Eq. (3) through the lens of MI. We begin by defining the optimal conditional denoiser, i.e., the minimum mean square error (MMSE) estimator of $\boldsymbol{\epsilon}$ from a noisy channel $\mathbf{z}_\gamma$, which mixes $\mathbf{z}$ and $\boldsymbol{\epsilon}$ at noise level $\gamma$, conditioned on slots $\mathbf{s}$:

$$\hat{\boldsymbol{\epsilon}}(\mathbf{z}_\gamma, \gamma, \mathbf{s}) = \mathbb{E}_{\boldsymbol{\epsilon} \sim p(\boldsymbol{\epsilon}|\mathbf{z}_\gamma, \mathbf{s})}[\boldsymbol{\epsilon}] = \underset{\tilde{\boldsymbol{\epsilon}}(\mathbf{z}_\gamma, \gamma, \mathbf{s})}{\arg\min} \, \mathbb{E}_{p(\boldsymbol{\epsilon})p(\mathbf{z}|\mathbf{s})} \left[ \| \boldsymbol{\epsilon} - \tilde{\boldsymbol{\epsilon}}(\mathbf{z}_\gamma, \gamma, \mathbf{s}) \|_2^2 \right] .$$

By approximating the regression problem with a neural network, we obtain an estimate of the MMSE denoiser, which coincides with the denoising objective of diffusion model training. Let $\tilde{\mathbf{s}}$ denote negative slots sampled from a distribution $q(\tilde{\mathbf{s}} \mid \mathbf{z})$. Under this setup, we state the following theorem.

**Theorem 1.** *Let $\mathbf{z}$ and $\mathbf{s}$ be two random variables, and let $\tilde{\mathbf{s}}$ denote a sample from a distribution $q(\tilde{\mathbf{s}} \mid \mathbf{z})$. Consider the diffusion process $\mathbf{z}_\gamma = \sqrt{\sigma(\gamma)}\mathbf{z} + \sqrt{\sigma(-\gamma)}\boldsymbol{\epsilon}$, with $\boldsymbol{\epsilon} \sim \mathcal{N}(0, \mathbf{I})$. Let $\hat{\boldsymbol{\epsilon}}(\mathbf{z}_\gamma, \gamma, \mathbf{s})$ denote the MMSE estimator of $\boldsymbol{\epsilon}$ given $(\mathbf{z}_\gamma, \gamma, \mathbf{s})$. Then the negative of mutual information (MI) between $\mathbf{z}$ and $\mathbf{s}$ admits the following form:*

$$-I(\mathbf{z}; \mathbf{s}) = \underbrace{\frac{1}{2} \int_{-\infty}^{\infty} \left( \mathbb{E}_{(\mathbf{z}, \mathbf{s}), \boldsymbol{\epsilon}} \left[ \| \boldsymbol{\epsilon} - \hat{\boldsymbol{\epsilon}}(\mathbf{z}_\gamma, \gamma, \mathbf{s}) \|^2 \right] - \mathbb{E}_{(\mathbf{z}, \tilde{\mathbf{s}}), \boldsymbol{\epsilon}} \left[ \| \boldsymbol{\epsilon} - \hat{\boldsymbol{\epsilon}}(\mathbf{z}_\gamma, \gamma, \tilde{\mathbf{s}}) \|^2 \right] \right) d\gamma}_{\Delta}$$
$$+ \mathbb{E}_{\mathbf{z}} \left[ D_{\text{KL}}(q(\tilde{\mathbf{s}} \mid \mathbf{z}) \| p(\tilde{\mathbf{s}} \mid \mathbf{z})) - D_{\text{KL}}(q(\tilde{\mathbf{s}} \mid \mathbf{z}) \| p(\tilde{\mathbf{s}})) \right] . \tag{4}$$

Direct optimization of Eq. (4) is infeasible, both due to the high sample complexity and the difficulty of evaluating the KL-divergence terms. The quantity $\Delta$ instead provides a practical handle: it measures the denoising gap between aligned and mismatched slots, and thus serves as a tractable

surrogate for MI. For this reason, we adopt the training objective in Eq. (3), which aligns directly with $\Delta$. Since the register slots $\bar{\mathbf{r}}$ are independent of the data, they do not influence $I(\mathbf{z}; \mathbf{s})$. Furthermore, when $\tilde{\mathbf{s}}$ are sampled independently of $\mathbf{z}$ such that $q(\tilde{\mathbf{s}} \mid \mathbf{z}) = p(\tilde{\mathbf{s}})$, the KL-divergence terms in Eq. (4) can be reinterpreted as dependency measures between $\mathbf{s}$ and $\mathbf{z}$:

**Corollary 1.** *With the additional assumption $q(\tilde{\mathbf{s}} \mid \mathbf{z}) = p(\tilde{\mathbf{s}})$ in Theorem 1, it follows that*

$$\Delta = -I(\mathbf{z}; \mathbf{s}) - D_{\mathrm{KL}}(p(\mathbf{z})p(\mathbf{s})||p(\mathbf{z}, \mathbf{s})). \tag{5}$$

Minimizing $\Delta$ therefore corresponds to maximizing MI plus an additional reverse KL-divergence in Eq. (5). Intuitively, this reverse KL-divergence contributes by rewarding configurations where the joint distribution $p(\mathbf{z}, \mathbf{s})$ and the product of marginals $p(\mathbf{z})p(\mathbf{s})$ disagree in the opposite direction of MI. In combination with the forward KL in MI, this enforces divergence in both directions, thereby promoting stronger statistical dependence between $\mathbf{z}$ and $\mathbf{s}$. Proofs are provided in Section A.

## 5 EXPERIMENTS

We design our experiments to address the following key questions: **(i)** How well does CODA perform on unsupervised object discovery across synthetic and real-world datasets? (Section 5.1) **(ii)** How effective are the learned slots for downstream tasks such as property prediction? (Section 5.2) **(iii)** Does CODA improve the visual generation quality of slot decoders? (Section 5.3) **(iv)** What is the contribution of each component in CODA? (Section 5.4) To answer these questions, CODA is compared against state-of-the-art fully unsupervised OCL methods, described in Section B.2.

**Datasets.** Our benchmark covers both synthetic and real-world settings. For synthetic experiments, we use two variants of the MOVi dataset (Greff et al., 2022): MOVi-C, which includes objects rendered over natural backgrounds, and MOVi-E, which includes more objects per scene, making it more challenging for OCL. For real-world experiments, we adopt PASCAL VOC 2012 (Everingham et al., 2010) and COCO 2017 (Lin et al., 2014), two standard benchmarks for object detection and segmentation. Both datasets substantially increase complexity compared to synthetic ones, due to their large number of foreground classes. VOC typically contains images with a single dominant object, while COCO includes more cluttered scenes with two or more objects. Further dataset and implementation details are provided in Section B.

### 5.1 OBJECT DISCOVERY

Object discovery evaluates how well slots bind to objects by predicting a set of masks that segment distinct objects in an image. Following prior works, we report the FG-ARI, a clustering similarity metric widely used in this setting. However, FG-ARI alone can be misleading, as it may favor either over-segmentation or under-segmentation (Kakogeorgiou et al., 2024; Wu et al., 2023; Seitzer et al., 2023), thus failing to fully capture segmentation quality. To provide a more comprehensive evaluation, we also report mean Intersection over Union (mIoU) and mean Best Overlap (mBO). Intuitively, FG-ARI reflects instance separation, while mBO measures alignment between predicted and ground-truth masks. On real-world datasets such as VOC and COCO, where semantic labels are available, we compute both mBO and mIoU at two levels: instance-level and class-level. Instance-level metrics assess whether objects of the same class are separated into distinct instances, whereas class-level metrics measure semantic grouping across categories. This dual evaluation reveals whether a model tends to prefer instance-based or semantic-based segmentations.

Table 1 shows results on synthetic datasets. CODA outperforms on both MOVi-C and MOVi-E. On MOVi-C, it improves FG-ARI by +7.15% and mIoU by +7.75% over the strongest baseline. On MOVi-E, which contains visually complex scenes, it improves FG-ARI by +2.59% and mIoU by +3.36%. In contrast, SLATE and LSD struggle to produce accurate object segmentations. Table 2 presents results on real-world datasets. CODA surpasses the best baseline (SlotAdapt) by +6.14% in FG-ARI on COCO. CODA improves instance-level object discovery by +3.88% $\mathrm{mBO}^i$ and +3.97% $\mathrm{mIoU}^i$, and semantic-level object discovery by +5.72% $\mathrm{mBO}^c$ and +7.00% $\mathrm{mIoU}^c$ on VOC. Qualitative results in Fig. 5 further illustrate the high-quality segmentation masks produced by CODA. Overall, these results demonstrate that CODA consistently outperforms diffusion-based OCL baselines by a significant margin. The improvements highlight its ability to obtain accurate segmentation, which facilitates compositional perception of complex scenes.

Table 1: Unsupervised object segmentation results on synthetic datasets. Results of other methods are reported from (Jiang et al., 2023; Akan & Yemez, 2025).

| MOVi-C | SLATE | SLATE$^+$ | LSD | Ours | MOVi-E | SLATE | SLATE$^+$ | LSD | SlotAdapt | Ours |
|---|---|---|---|---|---|---|---|---|---|---|
| mBO ($\uparrow$) | 39.37 | 38.17 | 45.57 | **46.55** | mBO ($\uparrow$) | 30.17 | 22.17 | 38.96 | **43.38** | 43.35 |
| mIoU ($\uparrow$) | 37.75 | 36.44 | 44.19 | **51.94** | mIoU ($\uparrow$) | 28.59 | 20.63 | 37.64 | 41.85 | **45.21** |
| FG-ARI ($\uparrow$) | 49.54 | 52.04 | 51.98 | **59.19** | FG-ARI ($\uparrow$) | 46.06 | 45.25 | 52.17 | 56.45 | **59.04** |

Table 2: Unsupervised object segmentation results on real-world datasets. † indicates results taken from (Wu et al., 2023), while results for other methods are taken from their respective papers.

| VOC | FG-ARI$\uparrow$ | mBO$^i\uparrow$ | mBO$^c\uparrow$ | mIoU$^i\uparrow$ | mIoU$^c\uparrow$ | COCO | FG-ARI$\uparrow$ | mBO$^i\uparrow$ | mBO$^c\uparrow$ | mIoU$^i\uparrow$ | mIoU$^c\uparrow$ |
|---|---|---|---|---|---|---|---|---|---|---|---|
| *MLP decoders* | | | | | | *MLP decoders* | | | | | |
| SA$^\dagger$ | 12.3 | 24.6 | 24.9 | - | - | SA$^\dagger$ | 21.4 | 17.2 | 19.2 | - | - |
| DINOSAUR | 24.6 | 39.5 | 40.9 | - | - | DINOSAUR | 40.5 | 27.7 | 30.9 | - | - |
| *Autoregressive decoders* | | | | | | *Autoregressive decoders* | | | | | |
| SLATE$^\dagger$ | 15.6 | 35.9 | 41.5 | - | - | SLATE$^\dagger$ | 32.5 | 29.1 | 33.6 | - | - |
| DINOSAUR | 24.8 | 44.0 | 51.2 | - | - | DINOSAUR | 34.1 | 31.6 | 39.7 | - | - |
| SPOT w/o ENS | 19.7 | 48.1 | 55.3 | 46.5 | - | SPOT w/o ENS | 37.8 | 34.7 | 44.3 | 32.7 | - |
| SPOT w/ ENS | 19.7 | 48.3 | 55.6 | 46.8 | - | SPOT w/ ENS | 37.8 | 35.0 | **44.7** | 33.0 | - |
| *Diffusion decoders* | | | | | | *Diffusion decoders* | | | | | |
| SlotDiffusion$^\dagger$ | 17.8 | 50.4 | 55.3 | 44.9 | 49.3 | SlotDiffusion$^\dagger$ | 37.2 | 31.0 | 35.0 | 31.2 | 36.5 |
| SlotAdapt | 29.6 | 51.5 | 51.9 | - | - | Stable-LSD | 35.0 | 30.4 | - | - | - |
| Ours | **32.23** | **55.38** | **61.32** | **50.77** | **56.30** | SlotAdapt | 41.4 | 35.1 | 39.2 | 36.1 | 41.4 |
| | | | | | | Ours | **47.54** | **36.61** | 41.43 | **36.41** | **42.60** |

## 5.2 PROPERTY PREDICTION

Following prior works (Dittadi et al., 2022; Locatello et al., 2020; Jiang et al., 2023), we evaluate the learned slot representations through downstream property prediction on the MOVi datasets. For each property, a separate prediction network is trained using the frozen slot representations as input. We employ a 2-layer MLP with a hidden dimension of 786 as the predictor, applied to both categorical and continuous properties. Cross-entropy loss is used for categorical properties, while mean squared error (MSE) is used for continuous ones. To assign object labels to slots, we use Hungarian matching between predicted slot masks and ground-truth foreground masks. This task evaluates whether slots encode object attributes in a disentangled and predictive manner, beyond simply segmenting objects.

We report classification accuracy for categorical properties (Category) and MSE for continuous properties (Position and 3D Bounding Box), , as shown in Table 3. With the exception of *3D Bounding Box*, CODA outperforms all baselines by a significant margin. The lower performance on 3D bounding box prediction is likely due to DINOv2 features, which lack fine-grained geometric details necessary for precise 3D localization. Overall, these results indicate that the slots learned by CODA capture more informative and disentangled object features, leading to stronger downstream prediction performance. This suggests that CODA encodes properties that enable controllable compositional scene generation.

Table 3: Representation quality. Mean squared error (MSE) is reported for spatial attributes, including 'Position' and '3D bounding box', while classification accuracy is reported for 'Category'. Results of other methods are taken from (Jiang et al., 2023; Akan & Yemez, 2025).

| MOVi-C | SLATE | SLATE$^+$ | LSD | Ours | MOVi-E | SLATE | SLATE$^+$ | LSD | SlotAdapt | Ours |
|---|---|---|---|---|---|---|---|---|---|---|
| Position ($\downarrow$) | 1.37 | 1.28 | 1.14 | **0.01** | Position ($\downarrow$) | 2.09 | 2.15 | 1.85 | 1.77 | **0.01** |
| 3D B-Box ($\downarrow$) | 1.48 | 1.44 | 1.44 | 2.11 | 3D B-Box ($\downarrow$) | 3.36 | 3.37 | **2.94** | 3.75 | 4.22 |
| Category ($\uparrow$) | 42.45 | 45.32 | 46.11 | **74.12** | Category ($\uparrow$) | 38.93 | 38.00 | 42.96 | 43.92 | **78.06** |

## 5.3 COMPOSITIONAL IMAGE GENERATION

To generate high-quality images, a model must not only encode objects faithfully into slots but also recombine them into novel configurations. We evaluate this capability through two tasks. First, we

assess *reconstruction*, which measures how accurately the model can recover the original input image. Second, we evaluate *compositional generation*, which tests whether slots can be recombined into new, unseen configurations. Following Wu et al. (2023), these configurations are created by randomly mixing slots within a batch. Both experiments are conducted on COCO. In our evaluation, we focus on image fidelity, since our primary goal is to verify that slot-based compositions yield visually coherent generations. We report Fréchet Inception Distance (FID) (Heusel et al., 2017) and Kernel Inception Distance (KID) (Bińkowski et al., 2018) as quantitative measures of image quality.

Table 4 shows that CODA outperforms both LSD and SlotDiffusion, and further achieves higher fidelity than SlotAdapt. In the more challenging compositional generation setting, it achieves the best results on both FID and KID, highlighting its effectiveness for slot-based composition. Beyond quantitative metrics, Figs. 3 and 12 demonstrates CODA's editing capabilities. By manipulating slots, the model can remove objects by discarding their corresponding slots or replace them by swapping slots across scenes. These examples highlight that CODA supports fine-grained, controllable edits in addition to faithful reconstructions. Overall, CODA not only preserves reconstruction quality but also significantly improves the ability of slot decoders to generalize compositionally, producing high-fidelity images even in unseen configurations.

Table 4: Image generation results for reconstruction and compositional generalization on the COCO dataset. Results of other methods are taken from (Akan & Yemez, 2025).

| Metric | Reconstruction | | | | Compositional generation | | | |
|---|---|---|---|---|---|---|---|---|
| | LSD | SlotDiffusion | SlotAdapt | Ours | LSD | SlotDiffusion | SlotAdapt | Ours |
| KID$\times 10^3$ | 19.09 | 5.85 | 0.39 | **0.35** | 103.48 | 57.31 | 34.38 | **30.44** |
| FID | 35.54 | 19.45 | 10.86 | **10.65** | 167.23 | 64.21 | 40.57 | **31.03** |

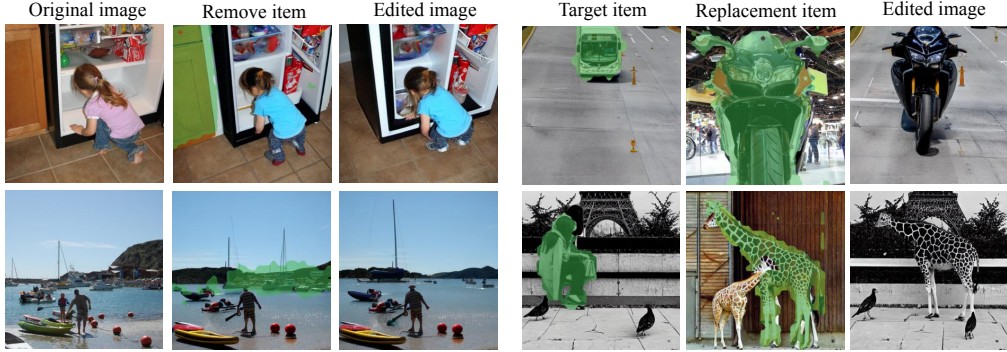

Figure 3: Illustration of compositional editing. CODA can compose novel scenes from real-world images by removing (left) or swapping (right) the slots, shown as masked regions in the images.

## 5.4 ABLATION STUDIES

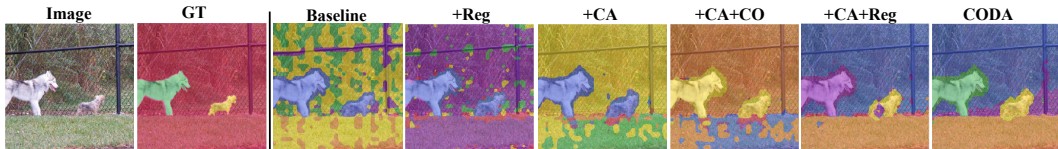

Figure 4: Illustration of the ablation study on VOC. We start from the pretrained diffusion model as a slot decoder (Baseline), adding register slots (Reg), finetuning the key, value, and output projections in the cross-attention layers (CA), adding contrastive alignment (CO).

We conduct ablations to evaluate the contribution of each component in our framework, with results on the VOC dataset summarized in Table 5. The baseline (first row) uses the frozen SD as a slot decoder. Finetuning the key, value, and output projections of the cross-attention layers (**CA**) yields moderate gains. Introducing register slots (**Reg**) provides substantial improvements, particularly in mBO, by

reducing slot entanglement. Adding the contrastive loss (**CO**) further boosts mIoU; however, applying it without stopping gradients in the diffusion model (∘) degrades performance. When combined, all components yield the best overall results, as shown in the final row, with qualitative examples in Fig. 4. Further ablation studies on the COCO dataset are reported in Table 9 and additional results are provided in Section E. Overall, the ablations demonstrate that each component contributes complementary benefits in enhancing compositional slot representations.

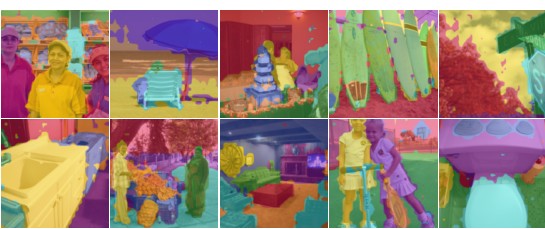

Figure 5: Segmentation masks learned by CODA on COCO

Table 5: Ablation study on the VOC dataset

| Component | | | Metric | | | | |
|---|---|---|---|---|---|---|---|
| **Reg** | **CA** | **CO** | FG-ARI↑ | mBO$^i$↑ | mBO$^c$↑ | mIoU$^i$↑ | mIoU$^c$↑ |
| | | | 12.27 | 47.21 | 54.20 | 48.72 | 55.71 |
| | ✓ | | 15.44 | 47.03 | 52.63 | 49.75 | 55.63 |
| ✓ | | | 19.21 | 55.76 | 64.02 | 49.93 | 57.14 |
| | | ✓ | 11.96 | 47.16 | 54.17 | 49.40 | 56.56 |
| ✓ | | ✓ | 19.62 | **56.27** | **65.05** | 50.40 | **58.02** |
| | ✓ | ✓ | 15.48 | 47.95 | 53.72 | **51.80** | 57.98 |
| ✓ | ✓ | | 31.27 | 54.30 | 59.44 | 50.62 | 55.63 |
| ✓ | ✓ | ∘ | 10.54 | 30.64 | 35.86 | 37.74 | 43.61 |
| ✓ | ✓ | ✓ | **32.23** | 55.38 | 61.32 | 50.77 | 56.30 |

## 6 CONCLUSIONS

We introduced CODA, a diffusion-based OCL framework that augments slot sequences with input-independent register slots and a contrastive alignment objective. Unlike prior approaches that rely solely on denoising losses or architectural biases, CODA explicitly encourages slot–image alignment, leading to stronger compositional generalization. Importantly, it requires no architectural modifications or external supervision, yet achieves strong performance across synthetic and real-world benchmarks, including COCO and VOC. Despite its current limitations (Section F), these results highlight the value of register slots and contrastive learning as powerful tools for advancing OCL.

## REPRODUCIBILITY STATEMENT

Section B.4 provides implementation details of CODA along with the hyperparameters used in our experiments. All datasets used in this work are publicly available and can be accessed through their official repositories. To ensure full reproducibility, the source code and pretrained checkpoints are available at https://github.com/sony/coda.

## LLM USAGE

In this work, large language models (LLMs) were used only to help with proofreading and enhancing the clarity of the text. All research ideas, theoretical developments, experiments, and implementation were conducted entirely by the authors.

## ETHICS STATEMENT

This work focuses on improving OCL and compositional image generation using pretrained diffusion models. While beneficial for controllable visual understanding, it carries risks: (i) misuse, as compositional generation could create misleading or harmful content; and (ii) bias propagation, since pretrained diffusion models may reflect biases in their training data, which can appear in generated images or representations. Our method is intended for research on OCL and representation, not for deployment in production systems without careful considerations.

### ACKNOWLEDGMENTS

The authors would like to thank Masato Ishii for helpful discussions on an earlier draft. We also thank anonymous reviewers for their constructive feedback and suggestions.

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

# Appendix

## A PROOFS

### A.1 PROOF OF THEOREM 1

To prove Theorem 1, we build on theoretical results that connect data distributions with optimal denoising regression. Let define the MMSE estimator of $\epsilon$ from a noisy channel $\mathbf{z}_\gamma$, which mixes $\mathbf{z}$ and $\epsilon$ at noise level $\gamma$ as

$$\hat{\epsilon}(\mathbf{z}_\gamma, \gamma) = \mathbb{E}_{\epsilon \sim p(\epsilon | \mathbf{z}_\gamma)}[\epsilon] = \underset{\tilde{\epsilon}(\mathbf{z}_\gamma, \gamma)}{\arg\min} \, \mathbb{E}_{p(\epsilon)p(\mathbf{z})}\left[\|\epsilon - \tilde{\epsilon}(\mathbf{z}_\gamma, \gamma)\|_2^2\right].$$

Kong et al. (2023) showed that the log-likelihood of $\mathbf{z}$ can be written solely in terms of the MMSE solution:

$$\log p(\mathbf{z}) = -\frac{1}{2} \int_{-\infty}^{\infty} \mathbb{E}_\epsilon\left[\|\epsilon - \hat{\epsilon}(\mathbf{z}_\gamma, \gamma)\|^2\right] d\gamma + c, \tag{6}$$

where $c = -\frac{D}{2}\log(2\pi e) + \frac{D}{2}\int_{-\infty}^{\infty}\sigma(\gamma)d\gamma$ is a constant independent of the data, with $D$ denoting the dimensionality of $\mathbf{z}$.

Analogously, defining the optimal denoiser $\hat{\epsilon}(\mathbf{z}_\gamma, \gamma, \mathbf{s})$ for the conditional distribution $p(\mathbf{z} \mid \mathbf{s})$ yields

$$\log p(\mathbf{z} \mid \mathbf{s}) = -\frac{1}{2} \int_{-\infty}^{\infty} \mathbb{E}_\epsilon\left[\|\epsilon - \hat{\epsilon}(\mathbf{z}_\gamma, \gamma, \mathbf{s})\|^2\right] d\gamma + c. \tag{7}$$

Let $\tilde{\mathbf{s}} \sim q(\tilde{\mathbf{s}} \mid \mathbf{z})$ denote slots sampled from an auxiliary distribution $q(\tilde{\mathbf{s}} \mid \mathbf{z})$, which may differ from $p(\tilde{\mathbf{s}} \mid \mathbf{z})$. Using the KL divergence, we obtain

$$
\begin{aligned}
D_{\mathrm{KL}}\left(q(\tilde{\mathbf{s}} \mid \mathbf{z})\|p(\tilde{\mathbf{s}} \mid \mathbf{z})\right) &= \mathbb{E}_{q(\tilde{\mathbf{s}}|\mathbf{z})}\left[\log \frac{q(\tilde{\mathbf{s}} \mid \mathbf{z})}{p(\tilde{\mathbf{s}} \mid \mathbf{z})}\right] \\
&= \mathbb{E}_{q(\tilde{\mathbf{s}}|\mathbf{z})}\left[\log q(\tilde{\mathbf{s}} \mid \mathbf{z}) - \log p(\mathbf{z} \mid \tilde{\mathbf{s}}) - \log p(\tilde{\mathbf{s}}) + \log p(\mathbf{z})\right] \\
&= -\mathbb{E}_{q(\tilde{\mathbf{s}}|\mathbf{z})}\left[\log p(\mathbf{z} \mid \tilde{\mathbf{s}})\right] + D_{\mathrm{KL}}(q(\tilde{\mathbf{s}} \mid \mathbf{z})\|p(\tilde{\mathbf{s}})) + \log p(\mathbf{z}).
\end{aligned}
$$

This leads to the following decomposition of the marginal distribution:

$$
\log p(\mathbf{z}) = \mathbb{E}_{q(\tilde{\mathbf{s}}|\mathbf{z})}[\log p(\mathbf{z} \mid \tilde{\mathbf{s}})] + D_{\mathrm{KL}}(q(\tilde{\mathbf{s}} \mid \mathbf{z})\|p(\tilde{\mathbf{s}} \mid \mathbf{z})) - D_{\mathrm{KL}}(q(\tilde{\mathbf{s}} \mid \mathbf{z})\|p(\tilde{\mathbf{s}}))
$$

Consequently, the mutual information (MI) between $\mathbf{z}$ and $\mathbf{s}$ can be expressed as

$$
\begin{aligned}
I(\mathbf{z}; \mathbf{s}) &= \mathbb{E}_{p(\mathbf{z},\mathbf{s})}\left[\log p(\mathbf{z} \mid \mathbf{s})\right] - \mathbb{E}_{p(\mathbf{z})}\left[\log p(\mathbf{z})\right] \\
&= \mathbb{E}_{p(\mathbf{z},\mathbf{s})}\left[\log p(\mathbf{z} \mid \mathbf{s})\right] - \mathbb{E}_{p(\mathbf{z})}\mathbb{E}_{q(\tilde{\mathbf{s}}|\mathbf{z})}[\log p(\mathbf{z} \mid \tilde{\mathbf{s}})] \\
&\quad - \mathbb{E}_{p(\mathbf{z})}[D_{\mathrm{KL}}(q(\tilde{\mathbf{s}} \mid \mathbf{z})\|p(\tilde{\mathbf{s}} \mid \mathbf{z})) - D_{\mathrm{KL}}(q(\tilde{\mathbf{s}} \mid \mathbf{z})\|p(\tilde{\mathbf{s}}))]
\end{aligned}
$$

From Eq. (7), it follows that

$$
\begin{aligned}
-I(\mathbf{z}; \mathbf{s}) &= \frac{1}{2}\int_{-\infty}^{\infty}\mathbb{E}_{(\mathbf{z},\mathbf{s}),\boldsymbol{\epsilon}}\left[\|\boldsymbol{\epsilon} - \hat{\boldsymbol{\epsilon}}(\mathbf{z}_\gamma, \gamma, \mathbf{s})\|^2\right]d\gamma - \frac{1}{2}\int_{-\infty}^{\infty}\mathbb{E}_{(\mathbf{z},\tilde{\mathbf{s}}),\boldsymbol{\epsilon}}\left[\|\boldsymbol{\epsilon} - \hat{\boldsymbol{\epsilon}}(\mathbf{z}_\gamma, \gamma, \tilde{\mathbf{s}})\|^2\right]d\gamma \\
&\quad + \mathbb{E}_{\mathbf{z}}\left[D_{\mathrm{KL}}(q(\tilde{\mathbf{s}} \mid \mathbf{z})\|p(\tilde{\mathbf{s}} \mid \mathbf{z})) - D_{\mathrm{KL}}(q(\tilde{\mathbf{s}} \mid \mathbf{z})\|p(\tilde{\mathbf{s}}))\right] \\
&= \frac{1}{2}\int_{-\infty}^{\infty}\left(\mathbb{E}_{(\mathbf{z},\mathbf{s}),\boldsymbol{\epsilon}}\left[\|\boldsymbol{\epsilon} - \hat{\boldsymbol{\epsilon}}(\mathbf{z}_\gamma, \gamma, \mathbf{s})\|^2\right] - \mathbb{E}_{(\mathbf{z},\tilde{\mathbf{s}}),\boldsymbol{\epsilon}}\left[\|\boldsymbol{\epsilon} - \hat{\boldsymbol{\epsilon}}(\mathbf{z}_\gamma, \gamma, \tilde{\mathbf{s}})\|^2\right]\right)d\gamma \\
&\quad + \mathbb{E}_{\mathbf{z}}\left[D_{\mathrm{KL}}(q(\tilde{\mathbf{s}} \mid \mathbf{z})\|p(\tilde{\mathbf{s}} \mid \mathbf{z})) - D_{\mathrm{KL}}(q(\tilde{\mathbf{s}} \mid \mathbf{z})\|p(\tilde{\mathbf{s}}))\right],
\end{aligned}
\tag{8}
$$

which completes the proof. $\qquad\square$

### A.2 Proof of Corollary 1

Under the assumption that $q(\tilde{\mathbf{s}} \mid \mathbf{z}) = p(\tilde{\mathbf{s}})$, it yields

$$
\begin{aligned}
\mathbb{E}_{\mathbf{z}}\left[D_{\mathrm{KL}}(q(\tilde{\mathbf{s}} \mid \mathbf{z})\|p(\tilde{\mathbf{s}}))\right] &= \mathbb{E}_{\mathbf{z}}\left[D_{\mathrm{KL}}(p(\tilde{\mathbf{s}})\|p(\tilde{\mathbf{s}}))\right] \\
&= 0.
\end{aligned}
\tag{9}
$$

Similarly, the expected KL-divergence term in Eq. (8) simplifies as follows:

$$
\begin{aligned}
\mathbb{E}_{\mathbf{z}}\left[D_{\mathrm{KL}}(q(\tilde{\mathbf{s}} \mid \mathbf{z})\|p(\tilde{\mathbf{s}} \mid \mathbf{z}))\right] &= \mathbb{E}_{\mathbf{z}}\left[D_{\mathrm{KL}}(p(\tilde{\mathbf{s}})\|p(\tilde{\mathbf{s}} \mid \mathbf{z}))\right] \\
&= \mathbb{E}_{p(\mathbf{z})p(\tilde{\mathbf{s}})}\left[\log \frac{p(\tilde{\mathbf{s}})}{p(\tilde{\mathbf{s}} \mid \mathbf{z})}\right] \\
&= \mathbb{E}_{p(\mathbf{z})p(\tilde{\mathbf{s}})}\left[\log \frac{p(\mathbf{z})p(\tilde{\mathbf{s}})}{p(\tilde{\mathbf{s}}, \mathbf{z})}\right] \\
&= D_{\mathrm{KL}}(p(\mathbf{z})p(\mathbf{s})\|p(\mathbf{z}, \mathbf{s})).
\end{aligned}
\tag{10}
$$

Substituting Eqs. (9) and (10) into Eq. (8) completes the proof. $\qquad\square$

**Remark 1.** *Eq. (10) shows that the additional expected KL-divergence reduces to the* reverse *KL divergence between the product of marginals $p(\mathbf{z})p(\mathbf{s})$ and the joint distribution $p(\mathbf{z}, \mathbf{s})$. This term complements the standard mutual information $I(\mathbf{z}; \mathbf{s})$, and together they form the Jeffreys divergence. Intuitively, while MI penalizes approximating the joint by the independent model, the reverse KL penalizes the opposite mismatch, thereby reinforcing the statistical dependence between $\mathbf{z}$ and $\mathbf{s}$.*

## B Experimental setup

This section outlines the experimental setup of our study. We detail the datasets, baseline methods, evaluation metrics, and implementation choices used in all experiments.

## B.1 DATASETS

**MOVi-C/E** (Greff et al., 2022). These two variants of the MOVi benchmark are generated with the Kubric simulator. Following prior works (Kakogeorgiou et al., 2024; Locatello et al., 2020; Seitzer et al., 2023), we evaluate on the 6,000-image validation set, since the official test sets are designed for out-of-distribution (OOD) evaluation. MOVi-C consists of complex objects and natural backgrounds, while MOVi-E includes scenes with a large numbers of objects (up to 23) per image.

**VOC** (Everingham et al., 2010). We use the PASCAL VOC 2012 "trainaug" split, which includes 10,582 images: 1,464 images from the official train set and 9,118 images from the SDB dataset (Hariharan et al., 2011). This configuration is consistent with prior works (Seitzer et al., 2023; Kakogeorgiou et al., 2024; Akan & Yemez, 2025). Training images are augmented with center cropping and then random horizontal flipping applied with a probability of 0.5. For evaluation, we use the official segmentation validation set of 1,449 images, where unlabeled pixels are excluded from scoring.

**COCO** (Lin et al., 2014). For experiments, we use the COCO 2017 dataset, consisting of 118,287 training images and 5,000 validation images. Training images are augmented with center cropping followed by random horizontal flipping with probability 0.5. For evaluation, we follow standard practice (Wu et al., 2023; Seitzer et al., 2023) by excluding crowd instance annotations and ignoring pixels corresponding to overlapping objects.

## B.2 BASELINES

We compare CODA against state-of-the-art fully unsupervised OCL models. The baselines include SA (Locatello et al., 2020), DINOSAUR (Seitzer et al., 2023), SLATE (Singh et al., 2022a), SLATE$^+$ (a variant using a pretrained VQGAN (Esser et al., 2021) instead of a dVAE), SPOT[2] (Kakogeorgiou et al., 2024), Stable-LSD[3] (Jiang et al., 2023) SlotDiffusion[4] (Wu et al., 2023), and SlotAdapt (Akan & Yemez, 2025). For DINOSAUR, we evaluate both MLP and autoregressive Transformer decoders. For SPOT, we report results with and without test-time permutation ensembling (SPOT w/ ENS, SPOT w/o ENS). We use the pretrained checkpoints released by the corresponding authors for SPOT and SlotDiffusion.

## B.3 METRICS

**Foreground Adjusted Rand Index (FG-ARI).** The Adjusted Rand Index (Hubert & Arabie, 1985) (ARI) measures the similarity between two partitions by counting pairs of pixels that are consistently grouped together (or apart) in both segmentations. The score is adjusted for chance, with values ranging from 0 (random grouping) to 1 (perfect agreement). The Foreground ARI (FG-ARI) is a variant that evaluates agreement only on foreground pixels, excluding background regions.

**Mean Intersection over Union (mIoU).** The Intersection over Union (IoU) between a predicted segmentation mask and its ground-truth counterpart is defined as the ratio of their intersection to their union. The mean IoU (mIoU) is obtained by averaging these IoU values across all objects and images in the dataset. This metric measures how well the predicted segmentation masks overlap with the ground-truth masks, aggregated over all instances.

**Mean Best Overlap (mBO).** The Best Overlap (BO) score for a predicted segmentation mask is defined as the maximum IoU between that predicted mask and any ground-truth object mask in the image. The mean BO (mBO) is then computed by averaging these BO scores across all predicted masks in the dataset. Unlike mIoU, which evaluates alignment with ground-truth objects directly, mBO emphasizes how well each predicted mask corresponds to its best-matching object, making it less sensitive to under- or over-segmentation.

---

[2] https://github.com/gkakogeorgiou/spot
[3] https://github.com/JindongJiang/latent-slot-diffusion
[4] https://github.com/Wuziyi616/SlotDiffusion

Table 6: Hyperparameters used for `CODA` on MOVi-C, MOVi-E, VOC, and COCO datasets

| Hyperparameter | MOVi-C | MOVi-E | VOC | COCO |
|---|---|---|---|---|
| **General** | | | | |
| Training steps | 250k | 250k | 250k | 500k |
| Learning rate | $2 \times 10^{-5}$ | $2 \times 10^{-5}$ | $2 \times 10^{-5}$ | $2 \times 10^{-5}$ |
| Batch size | 32 | 32 | 32 | 32 |
| Learning rate warm up | 2500 | 2500 | 2500 | 2500 |
| Optimizer | AdamW | AdamW | AdamW | AdamW |
| ViT architecture | DINOv2 ViT-B | DINOv2 ViT-B | DINOv2 ViT-B | DINOv2 ViT-B |
| Diffusion | SD v.1.5 | SD v.1.5 | SD v.1.5 | SD v.1.5 |
| Gradient norm clipping | 1 | 1 | 1 | 1 |
| Weighting $\lambda_{\mathrm{cl}}$ | 0.05 | 0.05 | 0.05 | 0.03 |
| **Image specification** | | | | |
| Image size | 512 | 512 | 512 | 512 |
| Augmentation | Rand.HFlip | Rand.HFlip | Rand.HFlip | Rand.HFlip |
| Crop | Full | Full | Central | Central |
| **Slot attention** | | | | |
| Input resolution | $32 \times 32$ | $32 \times 32$ | $32 \times 32$ | $32 \times 32$ |
| Number of slots | 11 | 24 | 6 | 7 |
| Number of iterations | 3 | 3 | 3 | 3 |
| Slot size | 768 | 768 | 768 | 768 |

## B.4 IMPLEMENTATION DETAILS

The hyperparameters are summarized in Table 6. We initialize the U-Net denoiser and VAE components from Stable Diffusion v1.5[5] (Rombach et al., 2022). During training, only the key, value, and output projections in the cross-attention layers are finetuned, while all other components remain frozen. For slot extraction, we employ DINOv2[6] (Oquab et al., 2024) with a ViT-B backbone and a patch size of 14, producing feature maps of size $32 \times 32$. The input resolution is set to $512 \times 512$ for the diffusion model and $448 \times 448$ for SA. As a form of data augmentation, we apply random horizontal flipping (Rand.HFlip) during training with a probability of 0.5. The negative slots are constructed by replacing 50% of the original slots with a subset of slots sampled from other images within the batch. `CODA` is trained using the Adam optimizer (Kingma & Ba, 2015) with a learning rate of $2 \times 10^{-5}$, a weight decay of 0.01, and a constant learning rate schedule with a warm-up of 2500 steps. To improve efficiency and stability, we use 16-bit mixed precision and gradient norm clipping at 1. All models are trained on 4 NVIDIA A100 GPUs with a local batch size of 32. We train for 500k steps on the COCO dataset and 250k steps on all other datasets. Training takes approximately 5.5 days for COCO and 2.7 days for the remaining datasets. For evaluation, the results are averaged over five random seeds. To ensure a fair comparision, for all FID and KID evaluations, we downsample CODA's $512 \times 512$ outputs to $256 \times 256$, matching the resolution used in prior works.

**Attention masks for evaluation.** We evaluate object segmentation using the attention masks produced by SA. At each slot iteration, attention scores are first computed using the standard `softmax` along the slot axis and then normalized via a weighted mean:

$$\mathbf{m}^{(t)} = \operatorname*{softmax}_{N} \left( \frac{q(\mathbf{s}^{(t)})k(\mathbf{f})^{\top}}{\sqrt{D}} \right) \implies \mathbf{m}^{(t)}_{m,n} = \frac{\mathbf{m}^{(t)}_{m,n}}{\sum_{l=1}^{M} \mathbf{m}^{(t)}_{l,n}},$$

where $D$ denotes the dimension of $k(\mathbf{f})$. The soft attention masks from the final iteration are converted to hard masks with `argmax` and used as the predicted segmentation masks for evaluation. This procedure ensures that each pixel is assigned to the slot receiving the highest attention weight.

---

[5] https://huggingface.co/stable-diffusion-v1-5/stable-diffusion-v1-5
[6] https://github.com/facebookresearch/dinov2

## C  VISUALIZATION OF ATTENTION SCORES

We visualize the attention scores in Fig. 6, showing the average attention mass assigned to semantic slots versus register slots. `CODA` is trained on the COCO dataset, and illustrative images are randomly sampled. Since the cross-attention layers in SD are multi-head, we average the attention maps across both heads and noise levels.

Interestingly, although register slots are semantically empty, they consistently absorb a substantial portion of the attention mass. This arises from the `softmax` normalization, which forces attention scores to sum to one across all slots. When a query does not strongly correspond to any semantic slot, the model must still allocate its attention; register slots act as neutral sinks that capture these residual values. This mechanism helps preserve clean associations between semantic slots and object concepts.

## D  COMPOSITIONAL IMAGE GENERATION FROM INDIVIDUAL SLOTS

We evaluate the ability of diffusion-based OCL methods to generate images from individual slots. As shown in Fig. 7, each input image is decomposed into six slots, with each slot intended to represent a distinct concept. We then condition the decoder on individual slots to generate single-concept images. The last column shows reconstructions using all slots combined. While all methods can reconstruct the original images when conditioned on the full slot set, most fail to produce faithful generations from individual slots. Specifically, Stable-LSD (Jiang et al., 2023) produces mostly texture-like patterns that poorly match the intended concepts, while SlotDiffusion (Wu et al., 2023), despite being trained end-to-end, also struggles to generate coherent objects. In Stable-LSD, slots are jointly trained to reconstruct the full scene, so object information can be distributed across multiple slots rather than concentrated in any single one. Consequently, removing all but one slot at test time puts the model in an out-of-distribution regime, and single-slot generations do not yield coherent objects even though the full slot set reconstructs the image well. This reflects slot entanglement where individual slots mix features from multiple objects. SlotAdapt (Akan & Yemez, 2025) partially alleviates this issue through an average register token, but since their embedding is injected directly into the cross-attention layers and tied to input-specific features, it limits flexibility in reusing slots across arbitrary compositions. In contrast, the input-independent register slots introduced in `CODA` act as residual sinks and do not encode input-specific features, enabling more faithful single-slot generations and greater compositional flexibility.

To quantify these results, we report FID and KID scores by comparing single-slot generations against the real images in the training set. For each validation image, we extract six slots and generate six corresponding single-slot images, ensuring a fair comparison across methods. Results on the VOC dataset are reported in Table 10, where `CODA` achieves the best scores, confirming its ability to generate coherent and semantically faithful images from individual slots.

Although register slots substantially reduce background entanglement, they do not enforce a hard separation between foreground and background. The attention mechanism in SA still remains soft, and our objectives do not explicitly prevent semantic slots from attending to background regions. As a result, semantic slots may still absorb contextual pixels, especially near object boundaries or in textured areas that are useful for reconstruction, when the number of slots exceeds the number of objects. As a result, small "meaningless" background fragments may still be assigned to semantic slots, as seen in Fig. 7. Empirically, however, we find that register slots substantially decrease background leakage compared to baselines without registers.

Table 7: Image generalization quality when using individual slots on the VOC dataset

| Metric | Stable-LSD | SlotDiffusion | SlotAdapt | Ours |
|---|---|---|---|---|
| KID$\times 10^3$ | 111.30 | 23.26 | 10.86 | **5.09** |
| FID | 189.77 | 94.88 | 47.70 | **27.61** |

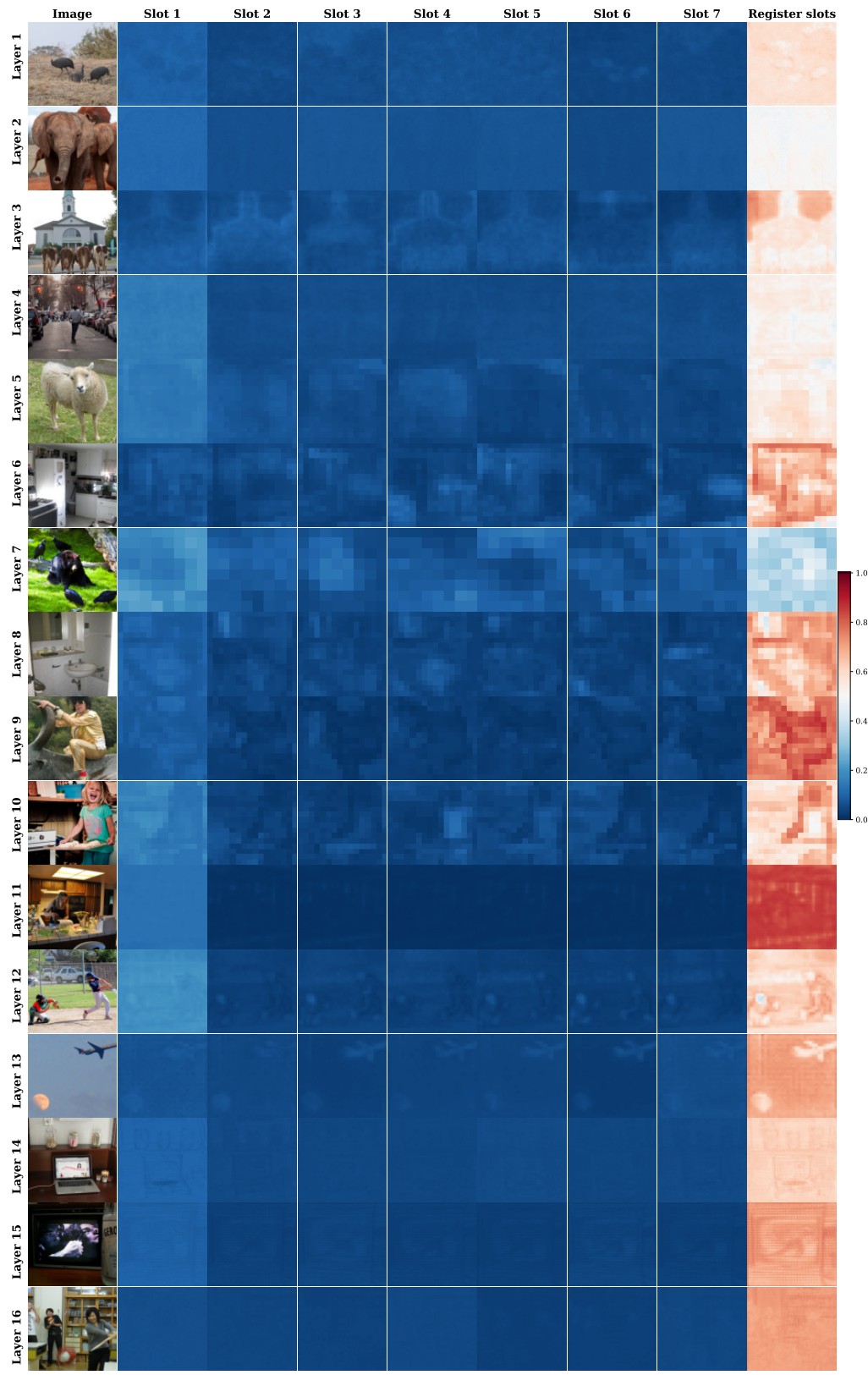

Figure 6: Attention scores across different cross-attention layers, averaged over heads and noise levels. The first column shows the original input image fed to CODA. Each image in row Layer $i$ and column Slot $j$ visualizes the total attention mass assigned to slot $j$ at layer $i$. The last column reports the total attention mass absorbed by the register slots. **CODA heavily attends to the register slots across all layers**.

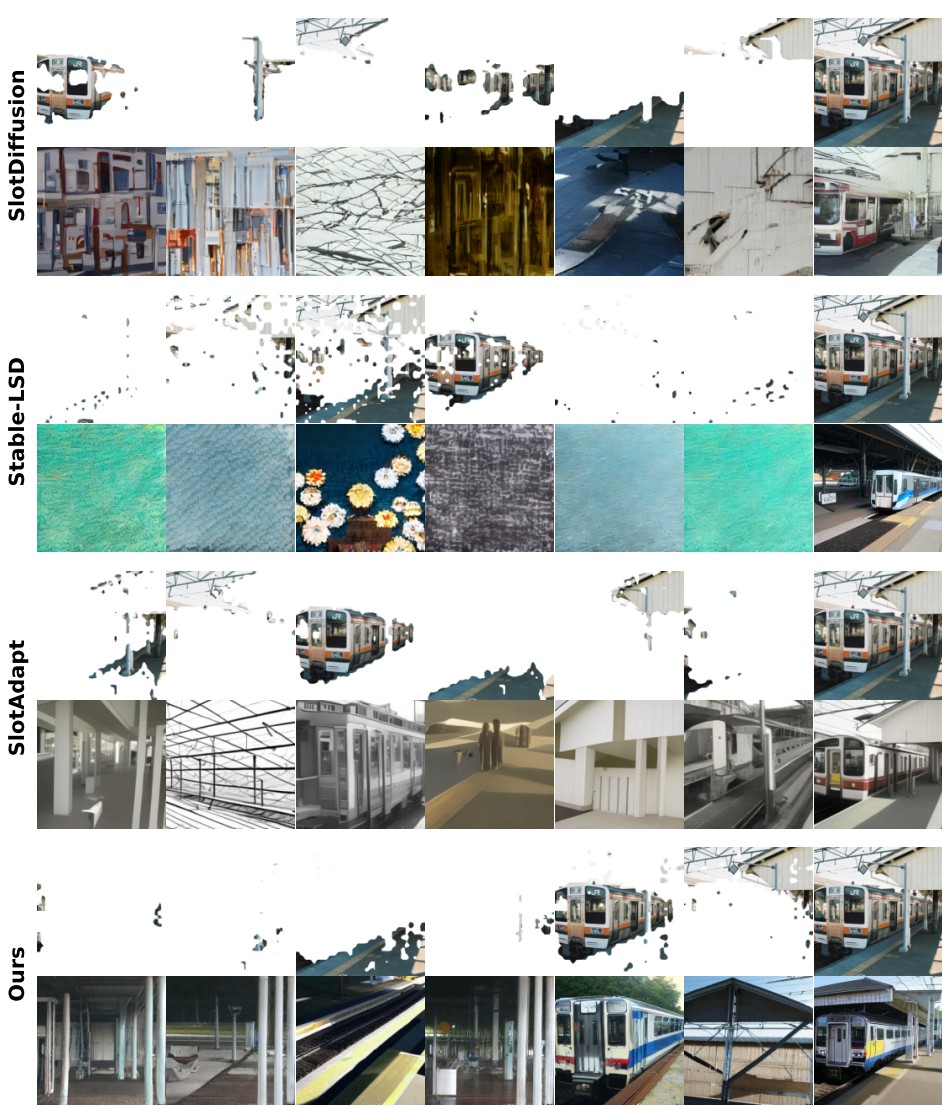

Figure 7: Image generation from individual slots. For each method, **Top:** slot masks, **Bottom:** generated images. The last column shows reconstructions from all slots. In CODA, register slots can be regarded as part of the U-Net architecture as they are independent from the input. Compared to baselines, our method generates faithful images from individual slots.

# E  ADDITIONAL RESULTS

In this section, we present supplementary quantitative and qualitative results that provide further insights into the performance of CODA.

## E.1  CLASSIFIER-FREE GUIDANCE

To enhance image generation quality, we employ classifier-free guidance (CFG) (Ho & Salimans, 2021), which interpolates between conditional and unconditional diffusion predictions. A guidance scale of CFG = 1 corresponds to standard conditional generation. We conduct an ablation study on different CFG values to assess their impact on generation quality. As shown in Fig. 8, both FID (Heusel et al., 2017) and KID (Bińkowski et al., 2018) scores improve with moderate guidance, with CODA achieving the best performance at CFG = 2.0. This indicates that a balanced level of guidance enhances fidelity without over-amplifying artifacts.

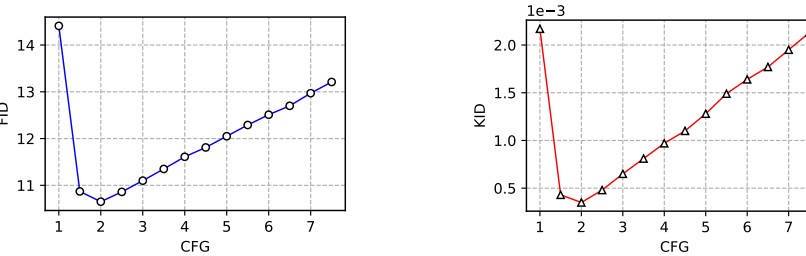

Figure 8: Generation fidelity on the COCO dataset for different CFG values

## E.2  LEARNABLE REGISTER SLOTS

Several works have explored trainable tokens as auxiliary inputs to transformers. For example, the [CLS] token is commonly introduced for classification in ViT (Dosovitskiy et al., 2021) and BERT (Devlin et al., 2019), while CLIP (Radford et al., 2021) employs an [EOS] token. These tokens serve as learnable registers that allow the model to store and retrieve intermediate information during inference. Goyal et al. (2024) demonstrated that appending such tokens can boost performance by increasing token interactions, thereby promoting deeper computation. Similarly, Darcet et al. (2024) utilized register tokens during pretraining to mitigate the emergence of high-norm artifacts. Motivated by these findings, we experiment with replacing our frozen CLIP-derived register slots with learnable ones. These slots are appended to the slot sequence but remain context-free placeholders.

Results on VOC with varying numbers of learnable register slots are shown in Table 8. The model without register slots ($R = 0$) performs the worst across all metrics (FG-ARI, mBO$^i$, mBO$^c$, mIoU$^i$, and mIoU$^c$). Interestingly, introducing just a single register slot leads to a significant performance boost. Further increasing the number of tokens to $R = 77$, matching the configuration used in CODA, yields only marginal improvements. Although more register slots could slightly increase computational cost, this is negligible as the number of register slots is relatively small. For instance, using 77 register slots increases GPU time by only 0.02% compared to the baseline without using any register slot. Interestingly, CODA achieves the best performance when using frozen register slots. These findings emphasize the effectiveness of register slots in improving the model performance.

Table 8: Ablation study on varying the number of register slots on the VOC dataset

| $R$ | FG-ARI↑ | mBO$^i$↑ | mBO$^c$↑ | mIoU$^i$↑ | mIoU$^c$↑ |
|---|---|---|---|---|---|
| 0 | 15.44 | 47.03 | 52.63 | 49.75 | 55.63 |
| 1 | 30.39 | 54.47 | 59.96 | 50.21 | 55.34 |
| 4 | 29.89 | 54.91 | 60.15 | 50.65 | 55.65 |
| 64 | 30.40 | 54.62 | 59.93 | 50.47 | 55.45 |
| 77 | 30.21 | 55.26 | 60.89 | **50.86** | 56.07 |
| CODA | **32.23** | **55.38** | **61.32** | 50.77 | **56.30** |

### E.3 ADDITIONAL ABLATION ON COCO

We further examine the contribution of frozen register slots on the COCO dataset, using pretrained SD as the slot decoder baseline. Different settings are evaluated: (i) adding register slots (+Reg), (ii) adding register slots combined with finetuning the key, value, and output projections in cross-attention layers (+CA), and adding the contrastive loss (+CO). As shown in Table 9, register slots consistently improve performance in both cases, demonstrating their robustness and effectiveness when integrated into the slot sequence.

Table 9: Ablation study on the COCO dataset

| Method | FG-ARI↑ | mBO$^i$↑ | mBO$^c$↑ | mIoU$^i$↑ | mIoU$^c$↑ |
|---|---|---|---|---|---|
| Baseline | 20.99 | 29.77 | 37.21 | 32.16 | 41.25 |
| Baseline + Reg | 23.64 | 31.14 | 39.07 | 32.64 | 41.91 |
| Baseline + CA | 36.99 | 33.82 | 38.08 | 35.04 | 41.24 |
| Baseline + CA + Reg | 45.95 | 35.80 | 40.32 | 35.76 | 41.75 |
| Baseline + CO | 25.24 | 30.14 | 38.77 | 32.83 | **42.99** |
| Baseline + CO + CA | 35.84 | 34.36 | 38.67 | 36.28 | 42.85 |
| Baseline + CA + Reg + CO (CODA) | **47.54** | **36.61** | **41.43** | **36.41** | 42.64 |

We further analyze the effect of the contrastive loss on image generation. Results are reported in Table 10. Without the contrastive loss, Reg+CA achieves slightly better FID/KID under compositional generation than the full model Reg+CA+CO. This aligns with the role of CO, which is primarily intended to strengthen slot–image alignment and object-centric representations rather than to maximize image fidelity, and can therefore marginally degrade FID/KID. Overall, CO should be viewed as an optional component that further improve object discovery at a small cost in visual quality.

Table 10: Image generation results for reconstruction and compositional generalization on the COCO dataset

| Metric | Reconstruction | | Compositional generation | |
|---|---|---|---|---|
| | Reg + CA | Reg + CA + CO | Reg + CA | Reg + CA + CO |
| KID$\times 10^3$ | 0.39 | **0.35** | **27.95** | 30.44 |
| FID | **10.65** | **10.65** | **29.34** | 31.03 |

### E.4 EFFECT OF THE CONTRASTIVE LOSS WEIGHTING

We conduct an ablation study to analyze the impact of the weighting coefficient $\lambda_{cl}$ in the contrastive loss term of our objective function in Eq. (3). Results on the COCO dataset are shown in Table 11. The study reveals that moderate values of $\lambda_{cl}$ achieve the best trade-off between the denoising and contrastive objectives, yielding the strongest overall performance. In practice, very small weights underuse the contrastive signal, while excessively large weights destabilize training and harm reconstruction quality. Although the contrastive loss shares a similar form with the diffusion loss, in practice, we find that it needs to be weighted by a relatively small factor $\lambda_{cl}$ to obtain good results. Empirically, increasing $\lambda_{cl}$ consistently degrades visual quality. We hypothesize that this happens because the contrastive term operates on slot-level features and, when heavily weighted, over-emphasizes alignment at the expense of the diffusion prior, leading to overspecialized and less realistic samples. In contrast, a small $\lambda_{cl}$ acts as a weak regularizer that improves alignment while keeping the diffusion objective dominant.

### E.5 COMBINATION RATIOS FOR NEGATIVE SLOTS

This section explores different combination ratios for constructing negative slots $\tilde{s}$. As outlined in Section 4.3, given two slot sequences $s$ and $s'$ from two distinct images $x$ and $x'$, we randomly replace a fraction $r \in (0, 1]$ of slots from $s$ with those from $s'$. When $r = 1$, the entire set of slots $s$

Table 11: Ablation study on varying the weighting terms in contrastive loss on the COCO dataset

| $\lambda_{\text{cl}}$ | FG-ARI↑ | mBO$^i$↑ | mBO$^c$↑ | mIoU$^i$↑ | mIoU$^c$↑ |
|---|---|---|---|---|---|
| 0 | 45.95 | 35.80 | 40.32 | 35.76 | 41.75 |
| 0.001 | 46.07 | 35.99 | 40.50 | 36.18 | 42.18 |
| 0.002 | 45.93 | 35.88 | 40.79 | 35.74 | 42.02 |
| 0.003 | **47.54** | **36.61** | **41.43** | **36.41** | **42.60** |
| 0.004 | 46.87 | 36.38 | 41.13 | 36.26 | 42.41 |
| 0.005 | 44.98 | 35.80 | 40.86 | 35.44 | 41.75 |

is replaced by $\mathbf{s}'$, while values $0 < r < 1$ yield mixed sets of slots $\tilde{\mathbf{s}}$ that only partially mismatch the original slots $\mathbf{s}$. Results on VOC (Table 12) show that $r = 0.5$ performs best, whereas $r = 1$ leads to overly trivial negative slots that provide little gradient signal. Intuitively, partial mismatches act as harder negatives, forcing the model to better discriminate correct slot-image alignments.

Table 12: Ablation study on varying the portion of negative slots on the VOC dataset

| $r$ | FG-ARI↑ | mBO$^i$↑ | mBO$^c$↑ | mIoU$^i$↑ | mIoU$^c$↑ |
|---|---|---|---|---|---|
| 0.25 | 32.34 | 54.58 | 60.52 | 50.44 | 55.97 |
| 0.50 | 32.23 | **55.38** | **61.32** | **50.77** | **56.30** |
| 0.75 | **33.34** | 55.06 | 60.98 | 50.12 | 55.61 |
| 1.00 | 32.67 | 54.60 | 59.84 | 49.73 | 54.64 |

### E.6 COMPARISON WITH WEAKLY-SUPERVISED BASELINES

We compare `CODA` to GLASS (Singh et al., 2025), a weakly supervised approach that uses a guidance module to produce semantic masks as pseudo ground truth. In particular, BLIP-2 (Li et al., 2023) is used for caption generation to create guidance signals. While this supervision helps GLASS mitigate over-segmentation, it also limits its applicability in fully unsupervised settings. In contrast, `CODA` does not rely on any external supervision and can distinguish between multiple instances of the same class, enabling more fine-grained object separation and richer compositional editing.

Table 13 reports the results. We additionally consider GLASS$^\dagger$, a variant of GLASS that uses ground-truth class labels associated with the input image. While GLASS achieves stronger performance on semantic segmentation masks, it underperforms `CODA` on object discovery, as reflected by lower FG-ARI scores. This suggests that `CODA` is better at disentangling distinct object instances at a conceptual level.

Table 13: Unsupervised object segmentation comparison with weakly-supervised OCL on real-world datasets, including VOC (left) and COCO (right). The results of GLASS and GLASS$^\dagger$ are taken from (Singh et al., 2025).

| VOC | FG-ARI↑ | mBO$^i$↑ | mBO$^c$↑ | mIoU$^i$↑ |
|---|---|---|---|---|
| GLASS$^\dagger$ | 21.3 | 58.5 | 61.5 | 57.8 |
| GLASS | 22.5 | **58.9** | **62.2** | **58.1** |
| Ours | **32.23** | 55.38 | 61.32 | 50.77 |

| COCO | FG-ARI↑ | mBO$^i$↑ | mBO$^c$↑ | mIoU$^i$↑ |
|---|---|---|---|---|
| GLASS$^\dagger$ | 32.5 | **40.8** | **48.7** | **39.0** |
| GLASS | 34.1 | 40.6 | 48.5 | 38.9 |
| Ours | **47.54** | 36.61 | 41.43 | 36.41 |

### E.7 QUALITATIVE COMPARISON

To complement the quantitative results in the main paper, we present additional qualitative examples that illustrate the effectiveness of `CODA`. These examples provide a more complete picture of the model's performance and highlight its advantages over previous approaches.

**Object segmentation.** We visualize segmentation results in Fig. 9. `CODA` consistently discovers objects and identifies semantically meaningful regions in a fully unsupervised manner. Compared to

diffusion-based OCL baselines such as Stable-LSD and SlotDiffusion, `CODA` produces cleaner masks with fewer fragmented segments, leading to more coherent object boundaries.

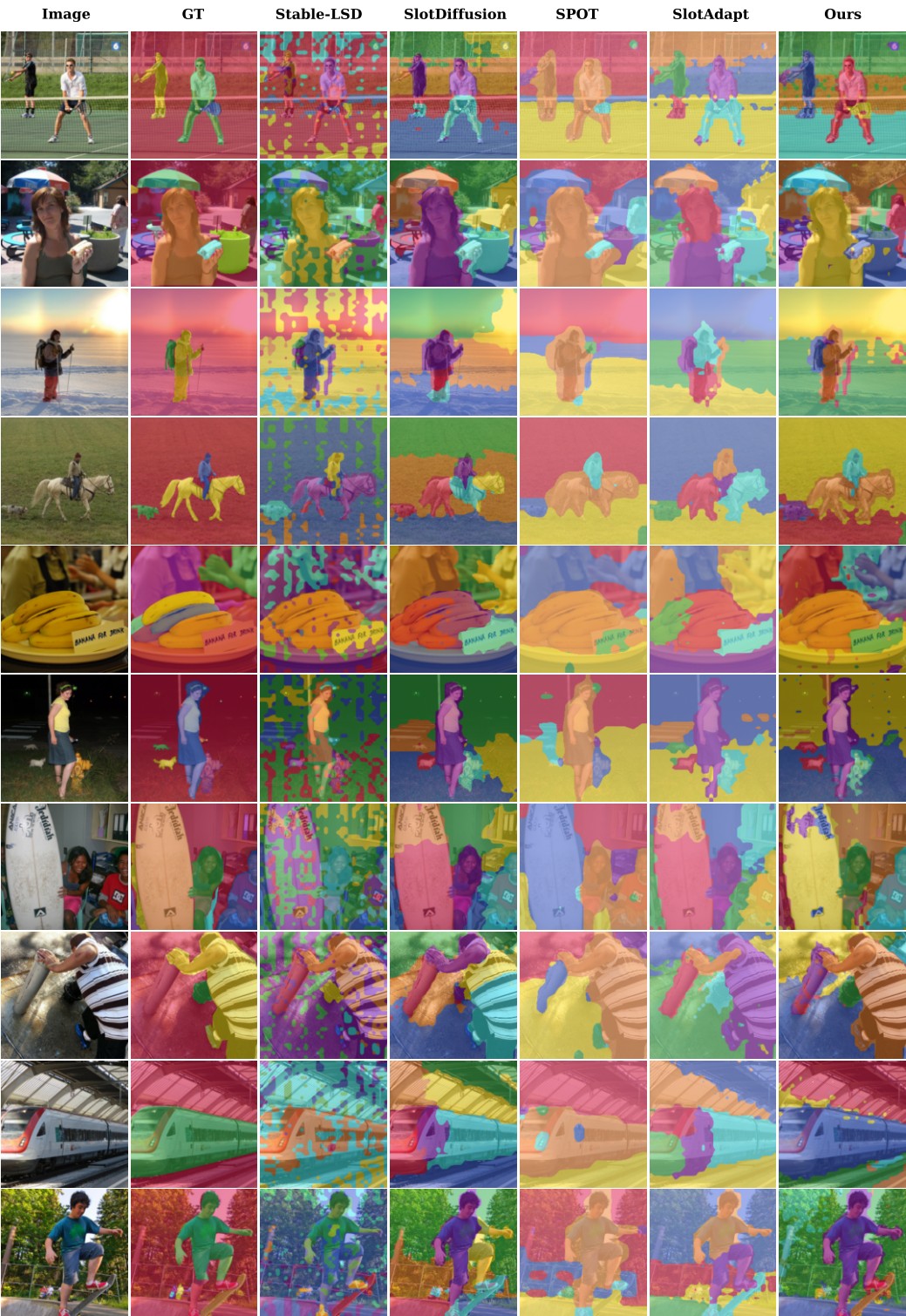

Figure 9: Visualization of image segmentation results on the COCO dataset. Compared to other methods, our method tends to produce more accurate masks with fewer fragmented segments.

**Reconstruction.** Figs. 10 and 11 show reconstructed images generated by `CODA`. The results demonstrate that `CODA` produces high-quality reconstructions when conditioned on the learned slots. Importantly, the generated images preserve semantic consistency while exhibiting visual diversity, indicating that the slots capture abstract and meaningful representations of the objects in the scene.

**Compositional generation.** Fig. 12 showcases COCO image edits based on `CODA`'s learned slots, including object removal, replacement, addition, and background modification. We find that the editing operations are highly successful, introducing only minor adjustments while consistently preserving high image quality.

## F LIMITATIONS AND FUTURE WORK

While `CODA` achieves strong performance across synthetic and real-world benchmarks, it has several limitations that open avenues for future research. (i) `CODA` relies on DINOv2 features and SD backbones, which may inherit dataset-specific biases and limit generalization to domains with very different visual statistics. (ii) While our contrastive loss improves slot–image alignment, full disentanglement in cluttered or ambiguous scenes remains an open challenge. (iii) Inherited from SA, `CODA` still requires the number of slots to be specified in advance. This restricts flexibility in scenes with a variable or unknown number of objects, and can lead to either unused slots or missed objects (Fan et al., 2024). In our implementation, `CODA` uses a fixed number of semantic slots plus a small number of register slots. Note that the register slots do not reduce semantic capacity but also cannot resolve the fundamental bottleneck when the true number of objects exceeds the available semantic slots, in which case objects may still be merged into the same slot despite reduced background entanglement. This is because the contrastive alignment is defined only for semantic slots, which encourages them to explain object-level content, while register slots are discouraged from encoding object-like structure.

Despite our improvements in object discovery and compositional control, faithfully preserving fine-grained images in reconstructions and compositional edits remains challenging, as also observed in prior slot diffusion models (e.g., SlotAdapt). We attribute this to several factors: (i) slot representations act as a low-dimensional bottleneck that must compress both geometry and detailed appearance; (ii) the diffusion backbone is pretrained to model images (and text–image pairs) but not to decode from slot-based object latents; and (iii) our training objective emphasizes object-centric grouping and controllability rather than exact pixel-level reconstruction. Improving image reconstruction in OCL is an important direction for future work.

Although CODA is conceptually compatible with a wide range of diffusion backbones, in this work we restrict ourselves to a relatively small, widely used backbone to ensure fair comparison with prior object-centric methods (e.g., SlotAdapt, LSD) and to keep computational and memory requirements manageable. We do not explore scaling CODA to larger, more recent architectures such as SDXL (Podell et al., 2024) or FLUX (Labs, 2024), which would require substantially more resources and additional engineering effort to handle larger feature maps, model sizes, and more sophisticated text-conditioning pipelines (e.g., multiple text encoders and auxiliary pooled text embeddings). Despite these limitations, we believe that `CODA` offers a scalable and conceptually simple foundation for advancing OCL. A promising direction for future work is extending `CODA` to Diffusion Transformers (DiTs) (Peebles & Xie, 2023), where slot representations could naturally replace or complement text embeddings in cross-attention, enabling richer and more flexible compositional control, as well as investigating integrations with larger backbones such as SDXL/FLUX to more fully assess the generality of our approach.

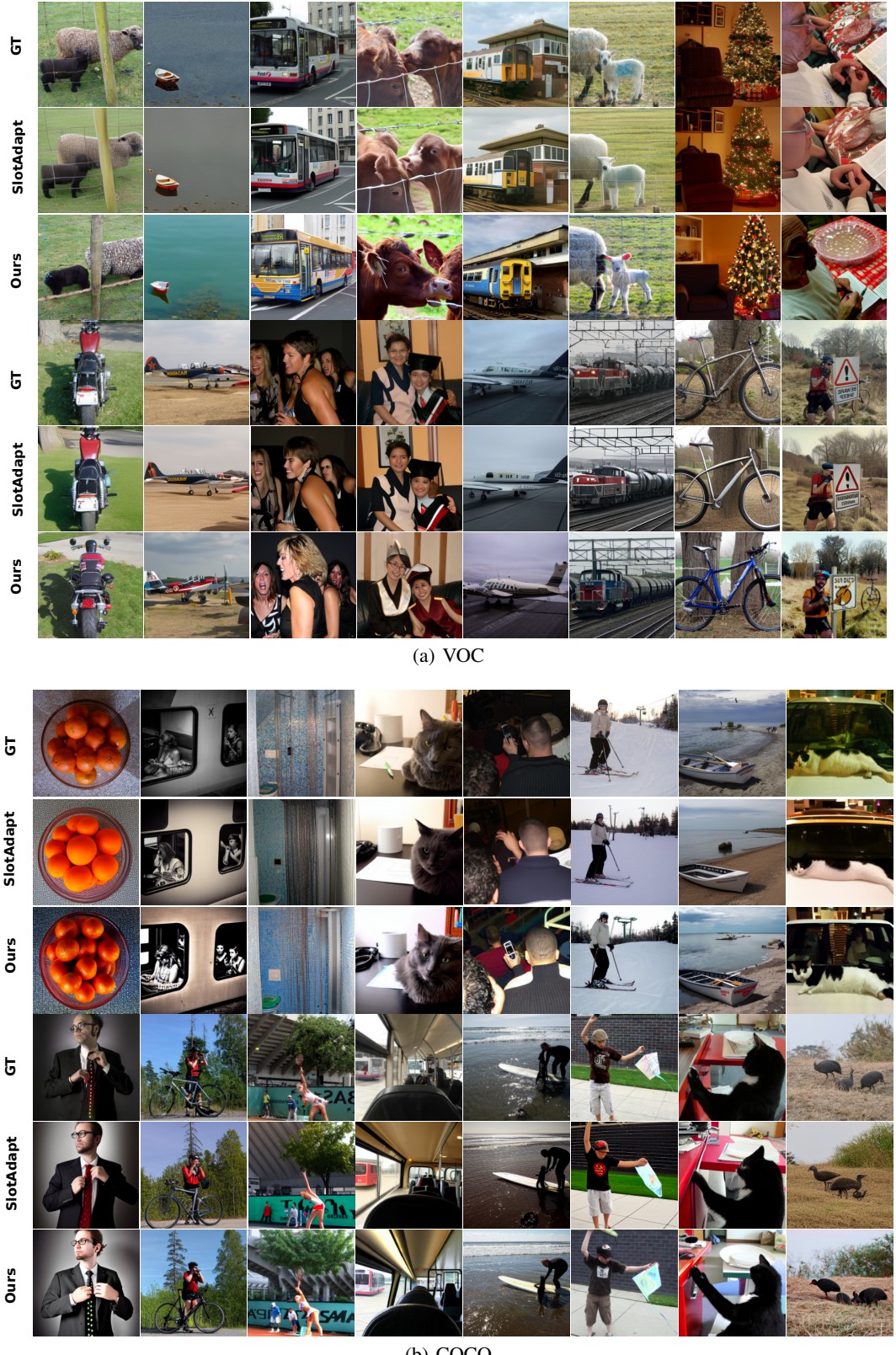

(a) VOC

(b) COCO

Figure 10: Reconstructed images on real-world datasets. **Top:** ground-truth (GT) images. **Middle:** images reconstructed by SlotAdapt. **Bottom:** images reconstructed by CODA.

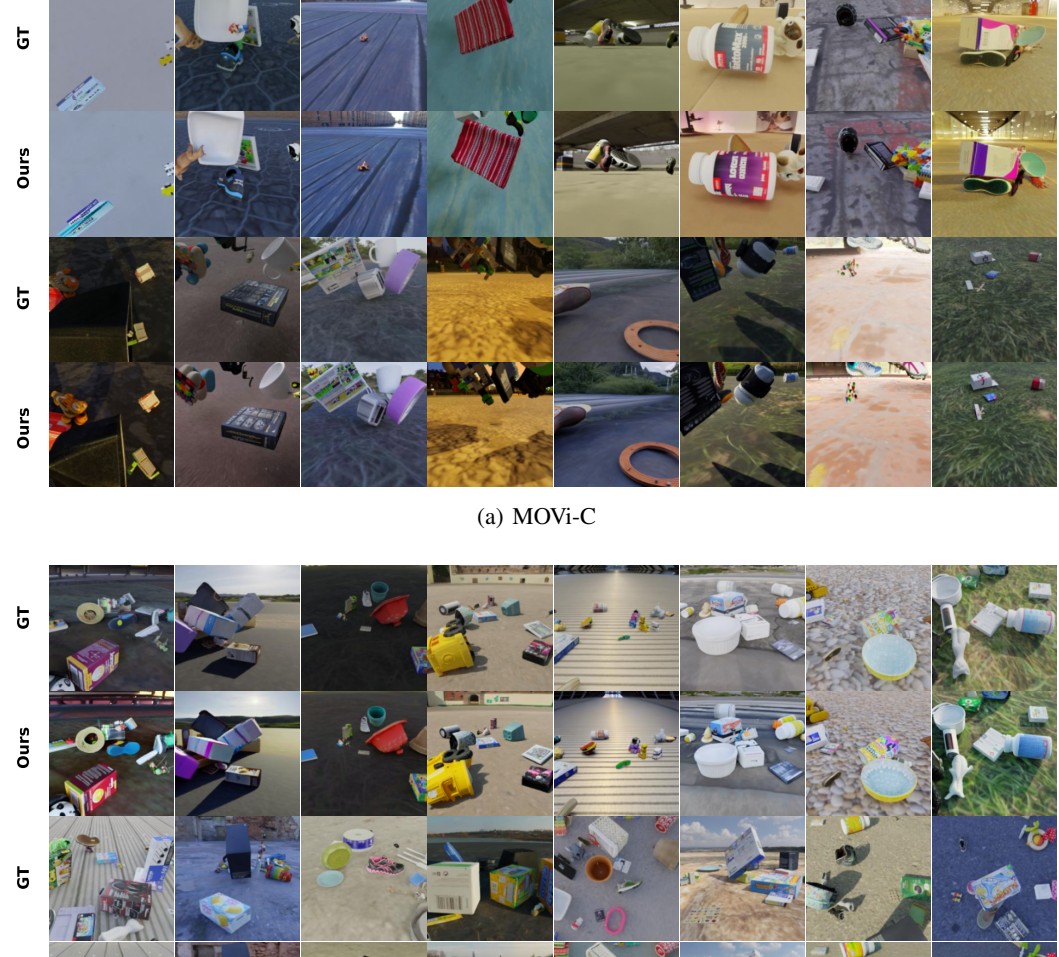

Figure 11: Reconstructed images on synthetic datasets. **Top:** ground-truth (GT) images. **Bottom:** images reconstructed by CODA.

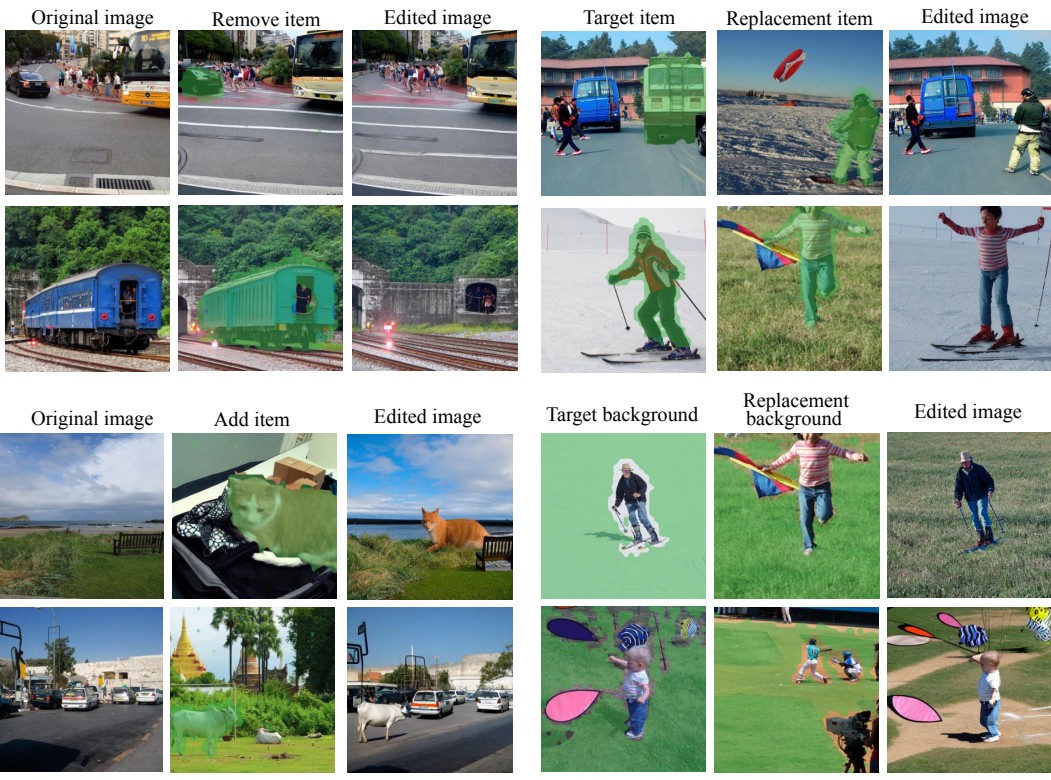

Figure 12: **Illustration of compositional editing.** CODA composes novel scenes from real-world images by removing (top left), swapping (top right), and adding (bottom left) slots, as well as changing the background (bottom right). The masked objects indicate the slots that are added, removed, or replaced relative to the original image.

