# OpenReview forum: "Improved Object-Centric Diffusion Learning with Registers and Contrastive Alignment"
_ICLR.cc/2026/Conference — ICLR 2026 Poster_

### Official Review · Reviewer_73eg · 2025-10-18

**Soundness:** 3
**Presentation:** 3
**Contribution:** 2
**Rating:** 4
**Confidence:** 5

**Summary:**

The paper presents a method for object-centric diffusion learning that extends Object-Centric Slot Diffusion (LSD) [1] by introducing two additional mechanisms: (i) register slots, designed to act as attention sinks, and (ii) a contrastive alignment loss to reduce slot entanglement. The model fine-tunes key and value projections in the cross-attention layers of a frozen Stable Diffusion backbone, while keeping the rest of the weights fixed.
The claimed novelty of register slots is, however, questionable. The concept was originally proposed in SlotAdapt [2], which introduced a separate learnable slot that serves exactly the same purpose. As also shown in Table 9 of the appendix, the authors acknowledge this similarity. The only difference is that in the current paper, the register slots are frozen rather than learnable, a minor technical modification rather than a conceptual contribution.
The proposed contrastive loss closely follows Learning to Compose [3], where compositional consistency is enforced through a shared slot initialization across samples. This paper adopts the same shared initialization trick, and the resulting contrastive objective is largely an adaptation rather than a new formulation.

[1] Jiang et al., LSD, NeurIPS 2023
[2] Akan & Yemez, SlotAdapt, ICLR 2025
[3] Jung et al., Learning to Compose, ICLR 2024

**Strengths:**

The method reports state-of-the-art results on both unsupervised object segmentation and compositional generation benchmarks, though the fairness of the compositional generation evaluation is debatable (see Weaknesses first and last point).
- The idea is conceptually simple, it does not introduce any architectural changes. (Although new register tokens can cost in terms of runtime (mostly negligible I guess, as the paper states that it only costs 0.02%) and contrastive loss is actually 2x forward (but in training))
- The experimental setup is solid, with comprehensive evaluations across multiple datasets and informative supplementary analyses in the appendix.

**Weaknesses:**

- The compositional generation evaluation is not entirely fair. The proposed method (CODA) is explicitly trained to reconstruct images given random slot combinations through its contrastive objective, which involves positive and negative slot pairs. This directly optimizes the model for composition-like reconstruction, so improved performance under this metric is somewhat expected. The qualitative results, however, are convincing and indicate stronger disentanglement than SlotAdapt, likely due to reduced slot entanglement from contrastive alignment.
- The paper’s claim that it "introduces" register slots is inaccurate. SlotAdapt (Akan & Yemez) already proposed a similar concept (see Table 9 in its appendix). While the current implementation with frozen register slots inspired by attention sinks in LLMs can be considered an improved version, it does not constitute a new contribution and should be framed as such.
- The contrastive loss significantly increases training cost, requiring 2× forward passes. 1x forward with 1/2 batch size is not possible because of the gradient stopping for the diffusion model.
- The ablation results are ambiguous. The combination of register slots and contrastive alignment (Reg+CA) already achieves nearly all the reported gains on COCO and VOC. The additional contrastive loss provides only marginal quantitative improvements despite its high computational cost. My guess is that this component mainly benefits compositional generation, since the contrastive objective explicitly encourages the model to reconstruct plausible images under slot mixing. If that is the case, it would be important to show both quantitative and qualitative results for compositional generation using the Reg+CA model (without CO) to confirm whether the improvement truly stems from better compositionality or merely from training bias toward the contrastive task.

**Questions:**

- The paper reports diffusion generation at 512×512 resolution, while most prior work (e.g., SlotAdapt, LSD) uses 256×256. How were comparisons conducted under this mismatch? Higher resolution can directly improve image quality metrics such as FID and KID.
- The training iterations are inconsistent across methods. SlotAdapt trains for 200K (MOVi-E), 250K (COCO), and 190K (VOC), while CODA uses 500K for COCO and 250K for VOC. Comparisons would be more meaningful if conducted under the same training budget.
- The paper would benefit from additional qualitative results on compositional editing to more clearly demonstrate improvements in visual plausibility and slot-level control.
- In Figure 7 (Appendix), it is unclear whether the single-slot generations include register slots. That is, whether the output is generated from one semantic slot plus registers or solely from the selected slot.
- On Table 3 results, is MOVi-E position prediction really correct? 0.01 looks like an anomaly.
- Results are reported for single runs, but given the stochastic nature of slot initialization, multi-seed evaluations would provide stronger statistical support.
- Since the proposed method is relatively lightweight, it could be scaled to larger backbones such as SDXL or FLUX variants to test its generality. No prior object-centric work appears to have explored this yet.
- An ablation on the classifier-free guidance (CFG) parameter, similar to that reported in SlotAdapt (Table 10, Appendix), would help analyze the effect of guidance strength.

Minor typo: Fig. 3 caption, "Orginal image" -> "Original image".

---

> ### Author Response · Authors · 2025-11-21
> **Response to Reviewer 73eg (1/4)**
>
> Thank you for your valuable comments. Please kindly find our responses below.
>
> ## Regarding contributions
> > The claimed novelty of register slots is, however, questionable. The concept was originally proposed in SlotAdapt [2]... The proposed contrastive loss closely follows Learning to Compose [3]
>
> Thank you for raising this concern. Although CODA is related to SlotAdapt and Learning-to-Compose, it differs from them in several key design aspects. For these reasons, we respectfully believe that our contributions go beyond an incremental change.
>
> - **Relation to SlotAdapt:** SlotAdapt’s register token is **context-dependent**: it is constructed inside SA (e.g., via an extra slot or by averaging semantic slots) and explicitly encodes global scene context, so it remains entangled with semantic content. In contrast, CODA’s register slots are **context-free**: they are input-independent, not derived from SA, and used solely as attention sinks. This design decouples registers from image-specific semantics and, as our experiments show, **makes them better suited for compositional generation** and slot reuse across images.
>
> - **Relation to Learning-to-Compose**: CODA’s contrastive loss enforces slot–image alignment by down-weighting the mismatched (negative) slot-image pairs and **does not directly optimize likelihood** over random slot mixtures. This is fundamentally different from Learning-To-Compose [3], which explicitly maximizes likelihood under random slot compositions and is directly tuned for composition-like reconstruction. In CODA, improvements in compositional generation arise **indirectly from better disentanglement and alignment**, not from training on the compositional generation task itself.
>
> - **Theoretical contribution**: To the best of our knowledge, we are the first to show that, when combined with the denoising loss, our training objective can be interpreted as a tractable surrogate for maximizing mutual information, providing a principled justification for our contrastive learning design.
>
> We have added explicit discussions of these connections and differences to the revised manuscript (Learning-to-Compose in Sec. 2, SlotAdapt in Sec. 4.1). Taken together, we believe these conceptual, theoretical, and empirical contributions go beyond a minor extension, and we kindly ask the reviewer to reconsider the assessment that CODA merely follows SlotAdapt or Learning-to-Compose.

---

> ### Author Response · Authors · 2025-11-21
> **Response to Reviewer 73eg (2/4)**
>
> ### [W1] Compositional generation evaluation
> > The compositional generation evaluation is not entirely fair. The proposed method (CODA) is explicitly trained to reconstruct images given random slot combinations through its contrastive objective, which involves positive and negative slot pairs. This directly optimizes the model for composition-like reconstruction, so improved performance under this metric is somewhat expected.
>
> We appreciate your comment but respectfully disagree that our compositional generation evaluation is not entirely fair. Our evaluation setup treats all methods equally:
>
> - **Same training data and supervision level**: All methods are trained on exactly the same images, with the same (fully unsupervised) supervision level, and sampling procedure. The only difference is the internal representation (register slots + contrastive alignment in CODA).
>
> - **No training on compositional images**: There is a misunderstanding in the statement from the reviewer that “CODA is explicitly trained to reconstruct images given random slot combinations through its contrastive objective”. Our contrastive loss operates on mixing slots and **is never given ground-truth images** formed from random cross-image slot mixtures. None of the methods (including CODA) is trained on ground-truth compositional images (e.g., swapping or adding slots across different images); such compositions are performed only at test time and are therefore **unknown for all models**.
>
> We therefore believe the compositional generation evaluation is fair, and kindly ask the reviewer to reconsider the assessment that CODA is unfairly trained for this metric.
>
> ### [W2] Contribution of register slots
> > The paper’s claim that it "introduces" register slots is inaccurate.
>
> We acknowledge that introducing additional tokens/slots as “registers” is not novel to our work, and we did not intend to claim otherwise. Our contribution is to extend and systematize this idea in the context of object-centric diffusion. We agree that the wording “introduces” was misleading and have
> - (i) rephrased our claim to state that CODA “employs” non-semantic register slots for object-centric diffusion, and
> - (ii) explicitly credited SlotAdapt for the use of a register token while clarifying the differences between their design and ours (see Sec. 4.1).
>
> Please also refer to our previous response regarding the differences between CODA and SlotAdapt.
>
> ### [W3] Training cost of CODA
> > The contrastive loss significantly increases training cost, requiring 2× forward passes. 1x forward with 1/2 batch size is not possible because of the gradient stopping for the diffusion model.
>
> Thank you for raising this point. We agree that the contrastive loss introduces computational overhead, as in our current implementation it requires two U-Net forward passes per training step for stability. However, **this extra cost is limited to training**: inference and sampling still use a single forward pass and are therefore unaffected. In addition, most of the backbone is frozen and shared across the two views, so the main overhead comes from the U-Net passes rather than duplicating all components. Given the improvements in object discovery and compositional generation shown in our ablations, we believe this additional training cost is justified.
>
> ### [W4] Ablation results
> > The ablation results are ambiguous. The combination of register slots and contrastive alignment (Reg+CA) already achieves nearly all the reported gains on COCO and VOC... it would be important to show both quantitative and qualitative results for compositional generation using the Reg+CA model (without CO)...
>
> Thank you for this careful analysis and for the suggestion to evaluate compositional generation with the Reg+CA variant (without the additional contrastive objective). To address your concern, we have run the requested experiment.
>
>
> |  Method         | rKIDx1000 | rFID   | gKIDx1000 | gFID   |
> | --------- | -------- | ----- | -------- | ----- |
> | Reg+CA    | 0.39     | **10.65** | **27.95**    | **29.34** |
> | Reg+CA+CO | **0.35**     | **10.65** | 30.44    | 31.03 |
>
> *(rFID and rKID indicate reconstruction results; gFID and gKID indicates generation results)*
>
> Interestingly, we find that Reg+CA achieves slightly better FID/KID under compositional generation than Reg+CA+CO. This is consistent with the fact that CO is primarily designed to strengthen slot–image alignment (i.e., object discovery and representation) rather than maximize image fidelity, and can therefore marginally degrade FID/KID. Therefore, the reviewer’s concern about the compositional evaluation being driven purely by “training bias toward the contrastive task” does not hold in our experiments.
>
> In the revised version, we’ve reported these additional results in Table 10 (Sec. E.3, Appendix).

---

> ### Author Response · Authors · 2025-11-21
> **Response to Reviewer 73eg (3/4)**
>
> ### [Q1] Diffusion generation at high resolution
> > The paper reports diffusion generation at 512×512 resolution, while most prior work (e.g., SlotAdapt, LSD) uses 256×256. How were comparisons conducted under this mismatch? Higher resolution can directly improve image quality metrics such as FID and KID.
>
> We thank the reviewer for pointing this out. To ensure a fair comparison, for all FID and KID evaluations we downsample CODA’s 512×512 outputs to 256×256, matching the resolution used by SlotAdapt and LSD. As shown in the table below, the FID and KID scores at 256×256 are even slightly better.
>
> | Resolution | rFID | rKIDx1000 | gFID | gKIDx1000
> | ---------- | ---------- |---------- |---------- | ---------------- |
> | 512x512    | 10.95          | **0.30**                     | 32.49 | 33.16 |
> | 256x256    | **10.65**          | 0.35                     | **31.03** | **30.44** |
>
> We have clarified this evaluation protocol in Appendix B.4 of the revised manuscript and updated the results (Tables 4, 7 and Fig. 8) accordingly.
>
> Please also refer to our general comment on `Clarification on the experimental setup` for segmentation results with 256x256.
>
> ### [Q2] Training iterations
> > The training iterations are inconsistent across methods...Comparisons would be more meaningful if conducted under the same training budget.
>
> Thank you for this comment. Our goal was to evaluate all methods under **their best-performing, recommended settings** rather than risk under- or over-training them with a uniform budget. Different frameworks have different numbers of trainable parameters and training setups, so **a fixed iteration cap would not translate to comparable convergence**. For CODA, we increased the number of iterations to ensure full convergence at the higher image resolution used in our setup. We do **not claim CODA to be more training-efficient** than prior works. For clarity, the table below summarizes the training budgets for several baseline methods, explicitly indicating that they do not share the same training budget.
>
> | Model         | MOVi-C      | MOVi-E           | VOC              | COCO              |
> | ------------- | ----------- | ---------------- | ---------------- | ----------------- |
> | DINOSAUR      | 500k (B=64) | 500k (B=64)      | 250k (B=64)      | 500k (B=64)       |
> | SlotAdapt     | N/A         | 200k (B=32)      | 190k (B=32)      | 250k (B=32)       |
> | SlotDiffusion | N/A         | 30 epochs (B=64) | 500 epoch (B=64) | 100 epochs (B=64) |
> | Stale-LSD     | 200k (B=64) | 200k (B=64)      | N/A              | N/A               |
> | CODA          | 250k (B=32) | 250k (B=32)      | 250k (B=32)      | 500k (B=32)       |
> *(B indicates the batch size. N/A indicates not available.)*
>
> ### [Q3] Additional qualitative results
> > The paper would benefit from additional qualitative results on compositional editing to more clearly demonstrate improvements in visual plausibility and slot-level control.
>
> Thank you for this suggestion. In the revised version (see Fig. 12), we’ve extended compositional edits on real scenes, including (i) removing objects, (ii) swapping objects between images, adding objects, and (iii) changing background. These additional visualizations more clearly demonstrate that CODA yields higher visual plausibility and more reliable slot-level control, consistent with our quantitative gains in object discovery and property prediction.
>
> ### [Q4] Clarification on single-slot generation
> > In Figure 7 (Appendix), it is unclear whether the single-slot generations include register slots.
>
> In Fig. 7, the “single-slot” generations are produced from one semantic slot together with the register slots. The register slots can be regarded as part of the U-Net architecture, since they are independent of the input.  Register slots are always present and serve only as non-semantic buffers that stabilize generation, i.e., they do not introduce additional object-level content on their own. We’ve clarified this in the caption to avoid confusion.
>
> ### [Q5] Results on MOVi-E
> > On Table 3 results, is MOVi-E position prediction really correct? 0.01 looks like an anomaly.
>
> We appreciate the reviewer pointing this out. For MOVi-E position prediction, we strictly followed the official evaluation script provided by LSD (https://github.com/JindongJiang/latent-slot-diffusion/blob/master/src/eval/eval.py). We have carefully re-checked the evaluation, re-running the script on our checkpoints, and confirmed that the reported 0.01 value is correct under this protocol. Position prediction itself is relatively straightforward; the large accuracy gap is likely due to the different visual backbones: CODA uses a frozen DINOv2 encoder, whereas both LSD and SlotAdapt employ a CNN encoder trained from scratch, which appears to be less effective for precise spatial localization on MOVi-E in our setting.

---

> ### Author Response · Authors · 2025-11-21
> **Response to Reviewer 73eg (4/4)**
>
> ### [Q6] Results with different runs
> > Results are reported for single runs, but given the stochastic nature of slot initialization, multi-seed evaluations would provide stronger statistical support.
>
> Thank you for this remark. We note that all reported numbers were already averaged over five random seeds. We have clarified this in the revised manuscript (see Appendix B.4). For completeness, we report below the mean ± standard deviation for the main results.
> | Dataset | FG-ARI         | mBOi          | mBOc         | mIoUi         | mIoUc        |
> | ------- | -------------- | ------------- | ------------ | ------------- | ------------ |
> | MOVi-C  | 59.19 (± 0.18) | 46.55 (±0.06) | NA           | 51.94 (±0.05) | NA           |
> | MOVi-E  | 59.04 (± 0.07) | 43.35 (±0.03) | NA           | 45.21 (±0.04) | NA           |
> | VOC     | 32.23 (± 0.19) | 55.38 (±0.10) | 61.32 (0.13) | 50.77 (±0.06) | 56.3 (±0.09) |
> | COCO    | 47.54 (± 0.20) | 36.61 (±0.10) | 41.43 (0.12) | 36.41 (±0.06) | 42.6 (±0.06) |
>
> ### [Q7] Extension of CODA to larger backbones
> >  Since the proposed method is relatively lightweight, it could be scaled to larger backbones such as SDXL or FLUX variants to test its generality.
>
> We appreciate this suggestion and agree that testing larger backbones is interesting. Conceptually, CODA is backbone-agnostic, so any gains from using larger backbones would be orthogonal to our contributions and would not change the core claims of the paper. In this work, we chose a smaller, widely used backbone to (i) enable fair comparison with prior object-centric methods (e.g., SlotAdapt, Stable-LSD), which are all implemented at this scale, and (ii) keep the computational and memory costs manageable.
>
> In particular, extending CODA to SDXL or FLUX-style backbones would require substantially more resources and non-trivial engineering to handle much larger feature maps and model sizes. This is because SDXL and FLUX use more sophisticated text-conditioning mechanisms (e.g., dual text encoders, pooled text embeddings as auxiliary conditions, or joint attention in FLUX), which would need careful integration with our framework.
>
> Given the strict time constraints of the ICLR rebuttal period, we are unable to conduct this extensive new experiment at this time. We see this as an exciting avenue for future work but beyond the scope of the current paper. We clarify this and add a brief discussion of such large-scale extensions in the limitation and future work section (Appendix F).
>
> ### [Q8] Ablation studies on classifier-free guidance
> > An ablation on the classifier-free guidance (CFG) parameter, similar to that reported in SlotAdapt (Table 10, Appendix), would help analyze the effect of guidance strength.
>
> We note that the ablation on classifier-free guidance was already included in Figure 8 of the manuscript. For completeness, we summarize the effects of varying the guidance strength below.
> | CFG | FID   | KIDx1000 |
> | --- | ----- | -------- |
> | 1   | 14.41 | 2.17     |
> | 1.5 | 10.87 | 0.43     |
> | 2   | **10.65** | **0.35**     |
> | 2.5 | 10.86 | 0.48     |
> | 3   | 11.1  | 0.65     |
> | 3.5 | 11.35 | 0.81     |
> | 4   | 11.61 | 0.97     |
> | 4.5 | 11.81 | 1.1      |
> | 5   | 12.05 | 1.28     |
> | 5.5 | 12.29 | 1.49     |
> | 6   | 12.51 | 1.64     |
> | 6.5 | 12.7  | 1.77     |
> | 7   | 12.97 | 1.95     |
> | 7.5 | 13.21 | 2.13     |
>
> CODA achieves the best performance at CFG=2.

---

> > ### Comment · Reviewer_73eg · 2025-11-21
> > **Post-rebuttal comments (1/2)**
> >
> > I thank the authors for the detailed rebuttal and for substantially extending the experiments and appendix. The new material clearly improves the paper: the added COCO ablations, the Reg+CA vs Reg+CA+CO comparison, the multi-seed reporting, the comparison to GLASS, the additional qualitative results, and the extra one-page supplementary with 256×256 training are all useful. The CODA-256 results in particular are important: they show that when CODA is trained at 256×256, it still improves FG-ARI over SlotAdapt on VOC and COCO, which alleviates part of my earlier concern about resolution differences.
> >
> > However, I still have several reservations:
> >
> > - **Contributions and relation to prior work.**
> >     Even with the improved discussion, I continue to see CODA as very close in spirit to existing SlotAdapt-style models and contrastive slot methods. SlotAdapt already introduces a global/register-like token to absorb non-object content; CODA’s shift to fixed, context-free CLIP pad embeddings plays a very similar architectural role (extra non-semantic slots acting as attention sinks) and looks more like a reparameterization and cleanup of that mechanism than a qualitatively new concept. Likewise, the contrastive loss is built around the same basic ingredients as Learning-to-Compose and related work (shared initialization, slot mixing, positive/negative slot configurations to encourage disentangled, composable slots), with a different formal objective and an MI-based interpretation. The theoretical analysis is interesting, but mainly provides a new perspective on a training signal that is already strongly inspired by existing formulations. Overall, I view CODA primarily as a strong engineering and design effort on top of SlotAdapt-style models rather than a fundamentally new object-centric diffusion framework.
> >
> > - **Backbone and training-budget fairness.**
> > The CODA-256 segmentation results help address the pure “512 vs 256” issue and demonstrate that CODA’s gains are not solely due to operating at 512×512. However, the main COCO comparisons still give CODA more training iterations than SlotAdapt (e.g., 500k vs 250k), and there are no matched-compute baselines or learning curves that separate gains from additional training from gains due to the proposed components. For older baselines such as SPOT and DINOSAUR, CODA further benefits from a stronger encoder (DINOv2) and different training setups. The revised paper is more transparent about these choices, which I appreciate, but I remain cautious in interpreting the headline SOTA results as purely method-driven.
> >
> > - **SD 1.5 at 256×256.**
> >   The rebuttal explains the choice of 512×512 by arguing that a largely frozen SD 1.5 backbone degrades significantly at substantially different resolutions. I agree that resolution mismatch can hurt performance when only cross-attention is adapted. At the same time, the SD 1.5 samples shown in the supplementary at 256×256 look unusually degraded compared to the 512×512 samples, and also compared to how SD 1.5 typically behaves at 256×256 in practice. This suggests that some aspects of the 256×256 pipeline (e.g., resizing, latent scaling, VAE usage, or other hyperparameters) may not be fully optimized. Because this is not discussed, I find it hard to take that figure alone as strong evidence that 256×256 generation is inherently “broken” in the frozen-backbone setting.
> >
> > - **Effect and cost of the contrastive loss.**
> > The new Reg+CA vs Reg+CA+CO experiment is very informative: it shows that most of the empirical gains come from the combination of register slots and cross-attention adaptation. The additional contrastive term provides relatively modest improvements in object-centric metrics, while requiring roughly twice as many diffusion forwards during training, and it slightly degrades compositional FID/KID. This is consistent with the authors’ claim that the contrastive term mainly targets slot–image alignment rather than image fidelity, but it also reinforces my impression that the “full CODA package” is only a small step beyond the simpler Reg+CA variant.

---

> ### Comment · Reviewer_73eg · 2025-11-21
> **Post-rebuttal comments (2/2)**
>
> - **Newly added compositional editing examples (Fig. 12).**
> I appreciate the newly added compositional editing results, but I do not find them as uniformly convincing as the text suggests. Several edits exhibit clear artifacts and incomplete control: for example, in the top-right example with the child in a striped T-shirt, the replacement is incomplete and the person is not cleanly replaced; in the bottom-left example, the added objects look visually inconsistent and unnatural. These examples show that some degree of compositional control is possible, but they also highlight that the edits are still fragile and far from reliably high visual plausibility.
>
> In summary, the authors have done a very good job strengthening the paper during the rebuttal, and CODA is a solid, empirically strong system. My remaining concerns are about how far the contributions go beyond existing SlotAdapt / Learning-to-Compose style ideas, and about how cleanly the reported gains can be attributed to the proposed components under matched backbone and training conditions, especially in light of the somewhat puzzling 256×256 behaviour and the still-fragile qualitative edits in Fig. 12.

---

> > ### Comment · Reviewer_vEFV · 2025-11-22
> > **Discussion on "Training Budget Fairness"**
> >
> > Dear Reviewer 73eg and authors,
> >
> > Sorry, but I hope you don't mind me jumping into this discussion. As another reviewer, I initially gave this paper a rating of 6 as I think it builds up a nice pipeline (or, engineering effort as you said) and the experimental results are pretty strong (especially the generation part).
> >
> > However, I didn't notice the training budget difference initially. Given that the authors actually "trained SlotAdapt with their code and verified that the resulting performance closely matches the numbers reported in the original paper" (see their response to my Q3), I guess a fair comparison here would be to train SlotAdapt longer for 500k steps and see if their performance gets better or is saturated.
> >
> > Best Regards,
> > Reviewer vEFV

---

> > > ### Author Response · Authors · 2025-11-25
> > > **Response to Reviewer vEFV**
> > >
> > > > I guess a fair comparison here would be to train SlotAdapt longer for 500k steps and see if their performance gets better or is saturated.
> > >
> > > We thank Reviewer `vEFV`  for this comment and we appreciate the additional perspective. Please refer to our general comments `Additional experiments with more training budget`.  The new results show that
> > > - SlotAdapt’s performance saturates with more iterations: we observe only marginal changes, and
> > > - Even with this extended training budget and higher resolution, SlotAdapt still does not close the gap to CODA
> > >
> > > These experiments indicate that CODA’s improvements cannot be explained by larger training  budget alone.

---

> > ### Author Response · Authors · 2025-11-25
> > **Response to Post-rebuttal comments (2/2) from Reviewer 73eg**
> >
> > > In summary, the authors have done a very good job strengthening the paper during the rebuttal, and CODA is a solid, empirically strong system. My remaining concerns are about how far the contributions go beyond existing SlotAdapt / Learning-to-Compose style ideas, and about how cleanly the reported gains can be attributed to the proposed components under matched backbone and training conditions, especially in light of the somewhat puzzling 256×256 behaviour and the still-fragile qualitative edits in Fig. 12.
> >
> > Thank you for acknowledging the revised version of the paper. We appreciate your comments and address your remaining concerns below.
> >
> > ### > How far CODA goes beyond prior works
> > We agree that CODA is related to both SlotAdapt and Learning-to-Compose and have made these connections explicit in our revised version. In our view, CODA goes beyond a pure engineering refinement in three concrete ways:
> >
> > - **New object-centric design**: CODA explicitly targets two core issues of existing slot diffusion models, namely **slot entanglement** and **weak slot–image alignment**, using *context-free register slots* and a *contrastive alignment* loss. This combination is not present in SlotAdapt or Learning-to-Compose and leads to substantially cleaner slots and more reliable single-slot generations (see Fig. 7 and Tab. 7).
> >
> > - **Simpler but more effective training objective:** Unlike Learning-to-Compose, CODA does not require an extra one-shot decoder or likelihood over random slot mixtures. The Learning-to-Compose setup substantially increases the number of learnable parameters, training cost, and implementation complexity (including approximate gradients for stability). In contrast, CODA’s loss yields a much simpler objective while still delivering clearly stronger object discovery, property prediction, and compositional generation.
> >
> > - **Theoretical grounding:** We show that, together with the denoising loss, CODA’s objective can be interpreted as a tractable surrogate for maximizing mutual information between slots and images, providing a principled justification for the design rather than a purely heuristic modification.
> >
> > In summary, we believe CODA offers a **simpler**, **theoretically grounded**, and **empirically stronger** object-centric diffusion framework, and that these aspects together go beyond a strong engineering effort and explain the substantial performance gains we observe.
> >
> >
> > ### > Qualitative edits
> > We agree that some of the edits in Fig. 12 are still not fully faithfull, especially on complex real images. Compositional editing with precise identity preservation and fidelity is inherently difficult, not only for CODA, but for diffusion-based slot OCL methods in general. CODA clearly improves disentanglement and slot-level control, but it does not yet guarantee perfectly stable edits or full identity preservation, particularly in difficult settings like changing backgrounds or in highly cluttered scenes. As discussed in Sec. F, we **explicitly acknowledge this as a limitation and view it as an important direction for future work**, e.g., by incorporating identity-aware objectives, better sampling techniques, etc.

---

> > > ### Comment · Reviewer_73eg · 2025-11-25
> > > **Additional post-rebuttal comment on contributions.**
> > >
> > > I have read the authors' latest reply arguing that CODA "goes beyond a pure engineering refinement" via (i) a new object-centric design, (ii) a simpler but more effective objective than Learning-to-Compose, and (iii) a theoretical mutual information grounding. I still disagree with this characterization.
> > >
> > > - **New object-centric design** The two main components the authors highlight, register-like slots and a contrastive alignment loss, are both already present in closely related work. SlotAdapt already uses a global/register token to absorb non-object content; CODA's context-free registers based on CLIP pad embeddings play essentially the same architectural role, with a different parameterization. On the loss side, CODA's contrastive term is built around shared initialization, slot mixing, and positive/negative slot configurations, which are exactly the core ingredients of Learning-to-Compose-style contrastive slot objectives. Combining these two ideas is reasonable, but I do not see this combination alone as a qualitatively new "object-centric design"; it is a reconfiguration of known mechanisms within the same SlotAdapt-style SD pipeline.
> > >
> > > - **Simpler but more effective objective** Removing an auxiliary decoder and likelihood term clearly simplifies the Learning-to-Compose setup, but simplifying an existing idea is not, by itself, a strong conceptual contribution. CODA's objective remains very close in spirit to existing slot-contrastive / LTC-like formulations, just with a lighter implementation. The main novelty here is practical (cleaner training loop, fewer parameters), not a new principle for learning compositional slots.
> > >
> > > - **Theoretical grounding** The mutual-information interpretation is a useful perspective, but it is essentially an analysis of a loss that is already heavily inspired by existing contrastive objectives. The theory does not change the algorithm or introduce qualitatively new behavior; it mostly justifies a design that is already very close to prior work. I do not find this sufficient to reclassify the contribution from "careful engineering and integration" to something more fundamental.
> > >
> > > I do want to acknowledge that the authors have clearly put a lot of effort into the rebuttal and additional experiments (extra training for SlotAdapt, 256×256 runs, more ablations, more qualitative results). These address most of my concerns about missing or unclear empirical evidence. My main remaining concern is conceptual: I still see CODA as a solid and useful system, but primarily as an incremental engineering refinement and integration of SlotAdapt / Learning-to-Compose–style ideas, rather than a clearly new framework.

---

> ### Author Response · Authors · 2025-11-25
> **Response to Post-rebuttal comments (1/2) from Reviewer 73eg**
>
> Thank you for the thoughtful follow-up and for acknowledging the expanded experiments and appendix. Below we would like to clarify your remaining concerns point by point.
>
> - **Backbone and training-budget fairness:** Please refer to our general comments `Additional experiments with more training budget` for a detailed discussion.
>
> - **Why the 256×256 samples of SD 1.5 in the supplement look worse than 512×512:** The supplementary figure was intentionally showing SD 1.5 run directly at 256×256 as a qualitative illustration of the **known resolution mismatch**. SD 1.5 was trained with a VAE and U-Net on 64×64 latents corresponding to 512×512 images; its weights were tuned to that resolution. Forcing the same model to operate on 32×32 latents (256×256) is out-of-distribution and, in our experience and in the official implementations, **consistently degrades quality unless the model is specifically retrained or fine-tuned for 256×256**. In our code we strictly follow the standard SD 1.5 pipeline (no unusual resizing, latent scaling, or VAE hacks), and we have double-checked that our 256×256 generations reproduce this expected degradation. They reflect a fundamental limitation of pretrained diffusion models [1,2,3]. This issue was also discussed by practitioners reported in diffusers github (see e.g., issue [#953](https://github.com/huggingface/diffusers/issues/953)).
>
>
> - **CODA at 256×256**: We note that CODA’s results at 256x256 were provided in the supplementary only for the reviewer reference. We would like to clarify that we **do not recommend training CODA at a different native resolution** than the pretrained backbone, especially since CODA finetunes only cross-attention layers. As expected, the performance of CODA was downgraded. Unlike CODA, SlotAdapt has substantially larger trainable parameters. This allows SlotAdapt to adapt well at different resolutions. In our work, CODA is intentionally designed to preserve and leverage the knowledge of the pretrained SD backbone. For this reason, we consider **CODA at 256×256 suboptimal and do not view this configuration as a meaningful basis for comparison**.
>
> - **Effect and cost of the contrastive loss:**  We agree that CODA's contrastive loss increases training cost, but our ablations (Tables 5 and 9) show that it also brings performance gains, making the additional computation worthwhile.
>
>     *Improvements of contrastive loss on VOC*
>     | Model | FG-ARI↑ | mBOi ↑ | mBOc ↑ | mIoUi ↑ | mIoUc ↑ |
>     | --- | --- | --- | --- | --- | --- |
>     | Reg+CA | 31.27 | 54.30 | 59.44 | 50.62 | 55.63 |
>     | CODA | **32.23** | **55.38** | **61.32** | **50.77** | **56.30** |
>
>     *Improvements of contrastive loss on COCO*
>     | Model | FG-ARI↑ | mBOi ↑ | mBOc ↑ | mIoUi ↑ | mIoUc ↑ |
>     | --- | --- | --- | --- | --- | --- |
>     | Reg+CA | 45.95 | 35.80 | 40.32 | 35.76 | 41.75 |
>     | CODA | **47.54** | **36.61** | **41.43** | **36.41** | **42.64** |
>
>
> #### References
> [1] Schusterbauer, Johannes, et al. "Diff2Flow: Training Flow Matching Models via Diffusion Model Alignment." CVPR. 2025.
>
> [2] He, Ruozhen, et al. "NoiseShift: Resolution-Aware Noise Recalibration for Better Low-Resolution Image Generation." arXiv preprint arXiv:2510.02307 (2025).
>
> [3] Cheng, Jiaxiang, et al. "Resadapter: Domain consistent resolution adapter for diffusion models." AAAI. Vol. 39. No. 3. 2025.

---

### Official Review · Reviewer_vEFV · 2025-10-24

**Soundness:** 3
**Presentation:** 3
**Contribution:** 2
**Rating:** 6
**Confidence:** 4

**Summary:**

This paper proposes Contrastive Object-centric Diffusion Alignment (CODA), an object-centric learning (OCL) framework with Slot-Attention and a pre-trained Stable Diffusion (SD) decoder. Prior work SlotAdapt already shows that using a powerful pre-trained SD model can greatly improve the generation quality of the OCL method. CODA further strengthens the results in three ways: 1) register slots (empty slots) to absorb attention on non-object areas to improve slot disentanglement, 2) an easier approach adapting the pre-trained text-conditioned cross-attention layer to condition on slots, and 3) contrastive alignment that maximizes the reconstruction loss when conditioned on unrelated slots. With these techniques, CODA outperforms baselines in both segmentation and generation on real-world datasets.

**Strengths:**

- In this era of foundation models, I always believe the community of OCL should move to more pre-trained models. This paper is a nice attempt at using DINO and SD models, and demonstrates effectiveness at large-scale real-world datasets.
- All three techniques in CODA are well-motivated and implemented. I appreciate the thorough ablations in the main paper and the Appendix. They help answer the effectiveness of each component very well.
- The analysis of mutual information (MI) is interesting and inspiring.

**Weaknesses:**

I don't see big weaknesses in the paper. Some minor weaknesses:
1. Each component is not very novel, as they pre-exist in other areas. Though I don't view this as a big issue -- combining them in a nice way and achieve strong results is also a good contribution.
2. Why not comparing with GLASS in the experiments? It seems that the segmentation results are similar to them (by checking tables in their paper), but I believe the generation capability of CODA must be stronger. Please include this in the paper.
3. I don't see enough qualitative generation comparison with SlotAdapt. While the quantitative results of CODA clearly outperform it, it's always beneficial to see some visual results. For example, I would like to see SlotAdapt results in Fig.9 and Fig.10 in the Appendix. Is it because they haven't released their code? There is a `Code` button on their webpage but it doesn't seem to lead to anywhere. Then how do you get SlotAdapt in Fig.7 of Appendix?

**Questions:**

See weaknesses. Some minor questions not affecting my decision:
1. Have you tried SD 2.1? In my experience, SD 2.1 has a similar model size / arch as SD 1.5, but often performs better in generation, likely due to its v-predicition formulation.
2. What do you think is still missing in achieving better reconstruction of input images from slots? Checking results in Fig.10, the reconstructed object's identity is still clearly not preserved. I understand this is also the case for SlotAdapt so I don't view it as a weakness of the paper.
3. In the qualitative segmentation results in Fig.7, why are there still some meaningless background pixels being segmented to a slot? E.g., column 1, 2 in the train example, they don't seem to correspond to any objects. In theory, they should be absorbed by the register slots?

---

> ### Author Response · Authors · 2025-11-21
> **Response to Reviewer vEFV (1/2)**
>
> We sincerely thank the reviewer for the thorough and constructive comments. Please kindly find our responses below.
>
> ### [Q1] Comparison with GLASS
> >  Why not comparing with GLASS in the experiments?  It seems that the segmentation results are similar to them (by checking tables in their paper), but I believe the generation capability of CODA must be stronger. Please include this in the paper.
>
> Thank you for this helpful suggestion. GLASS is indeed a strong and closely related baseline, and we agree that a comparison is informative. In the revised version, we have added a detailed comparison to GLASS (see Appendix E.6. (Table 12).  GLASS is weakly supervised and relies on a guidance module to produce semantic masks as pseudo ground truth. While GLASS attains higher scores on semantic segmentation masks, it underperforms CODA on object discovery metrics (FG-ARI), indicating that CODA better disentangles individual object instances. Importantly, CODA is fully unsupervised and supports compositional image generation and editing at the slot level.
>
> ### [Q2] Qualitative generation comparison with SlotAdapt
> > I don't see enough qualitative generation comparison with SlotAdapt. While the quantitative results of CODA clearly outperform it, it's always beneficial to see some visual results.
>
> We thank the reviewer for this suggestion and agree that qualitative comparisons with SlotAdapt are valuable in addition to the quantitative results. In the revised manuscript, we’ve added corresponding rows with SlotAdapt generations to these figures to provide a more direct visual comparison (see Fig. 9). Compared to SlotAdapt, CODA tends to produce more accurate masks with fewer fragmented segments.
>
> In addition, we’ve included more qualitative CODA results for compositional generation (see Fig. 12), further illustrating the strength of our approach.
>
> ### [Q3] Implementation of SlotAdapt
> > There is a Code button on their webpage but it doesn't seem to lead to anywhere. Then how do you get SlotAdapt in Fig.7 of Appendix?
>
> To obtain SlotAdapt results, we used the official implementation provided by the authors as supplementary material https://openreview.net/notes/edits/attachment?id=V0Agy9L2XN&name=supplementary_material. We trained SlotAdapt with their code and verified that the resulting performance closely matches the numbers reported in the original paper.
>
> ### [Q4] Experiments with SD 2.1
> > Have you tried SD 2.1? In my experience, SD 2.1 has a similar model size / arch as SD 1.5, but often performs better in generation, likely due to its v-predicition formulation.
>
> We appreciate this suggestion. We've done some initial experiments using SD 2.1 as the backbone, keeping our architecture and training setup otherwise identical. In our setting, however, SD 2.1 consistently underperformed SD 1.5, in terms of segmentation metrics.
>
> | Dataset | Backbone | FG-ARI↑ | mBOi ↑ | mBOc ↑ | mIoUi ↑ | mIoUc ↑ |
> | --- | --- | --- | --- | --- | --- | --- |
> | VOC | SD1.5 | **32.23** | **55.38** | **61.32** | **50.77** | **56.30** |
> |   | SD2.1 | 27.51 | 53.22 | 54.46 | 50.19 | 52.07 |
> | COCO | SD1.5 | **47.54** | **36.61** | **41.43** | **36.41** | **42.60** |
> |   | SD2.1 | 44.12 | 32.31 | 35.54 | 32.66 | 37.32 |
>
> We suspect this is due to a mismatch between SD 2.1’s training resolution 768x768 and ours 512x512 since SD 2.1 was designed to operate the best at 768x768. We believe that with more careful selection of hyperparameter the perormance of SD 2.1 can be improved.
>
> ### [Q5] Image reconstruction quality
> >  What do you think is still missing in achieving better reconstruction of input images from slots? Checking results in Fig.10, the reconstructed object's identity is still clearly not preserved.
>
> We agree that faithfully preserving input images from slots remains an open challenge, not only for CODA but also for prior slot diffusion models such as SlotAdapt. In our view, there are several factors that currently limit reconstruction quality:
> - Slot representations are relatively low-dimensional bottlenecks that must compress both geometry and fine-grained appearance.
> - The diffusion backbone is pretrained to model images (and text–image pairs) but not to decode from slot-based object latents.
> - Our objective focuses on object discovery and compositional control rather than exact pixel-level reconstruction, so some loss of identity is tolerated in favor of a more robust slot–image alignment.
>
> We see improving reconstruction quality as an exciting direction for future work: richer slot parameterizations, additional reconstruction losses (e.g., perceptual loss), and incorporating additional supervision when available. In the revised manuscript, we’ve added a short discussion of these points (see Sec. F).

---

> ### Author Response · Authors · 2025-11-21
> **Response to Reviewer vEFV (2/2)**
>
> ### [Q6] Qualitative segmentation results
> > In the qualitative segmentation results in Fig.7, why are there still some meaningless background pixels being segmented to a slot? E.g., column 1, 2 in the train example, they don't seem to correspond to any objects. In theory, they should be absorbed by the register slots?
>
> Thank you for this careful observation. In our design, register slots are intended to absorb most of the background and ambiguous regions, but they do not enforce a hard separation between “background” and “semantic” slots. The attention mechanism in SA still remains soft, and our objectives do not explicitly prevent semantic slots from attending to background regions. As a result, semantic slots may still absorb contextual pixels, especially near object boundaries or in textured areas that are useful for reconstruction, when the number of slots exceeds the number of objects. As a result, small “meaningless” background fragments may still be assigned to semantic slots, as seen in Figure 7. Empirically, however, we find that register slots substantially decrease background leakage compared to baselines without registers.
>
> We’ve clarified this point in the text and added a brief discussion of residual background leakage in Appendix (Sec. D).

---

> > ### Comment · Reviewer_vEFV · 2025-11-22
> >
> > I thank the reviewer for the rebuttal. Most of my concerns are addressed. Regarding Q2 though, I still want to see SlotAdapt's results in Fig. 10 as it is a direct comparison of reconstruction capability between SlotAdapt and CODA. I assume it's too hard to add since you have their model.

---

> > > ### Author Response · Authors · 2025-11-25
> > > **Reconstruction results of SlotAdapt in Fig. 10**
> > >
> > > We thank the reviewer for the constructive suggestions to strengthen the paper. We had indeed overlooked this in the initial revision. The updated version now includes SlotAdapt’s results in Fig. 10, providing a direct visual comparison of reconstruction capability between SlotAdapt and CODA.

---

> ### Author Response · Authors · 2025-11-28
>
> We greatly appreciate the reviewer’s suggestion and constructive feedback. If any further clarification is needed, we would be happy to provide it.

---

### Official Review · Reviewer_hjxk · 2025-11-01

**Soundness:** 3
**Presentation:** 4
**Contribution:** 3
**Rating:** 8
**Confidence:** 4

**Summary:**

The paper addresses the issue of binding of objects to slots in existing slot-based OCL methods. It proposes using register tokens/slots and a contrastive loss to alleviate this issue, and also fine-tunes the query/key embeddings in the diffusion model to better adapt it to the slot representation. The effectiveness of the proposed solution is evaluated on standard datasets in terms of various tasks, including object discovery, property prediction, and compositional generation. Furthermore, detailed ablations demonstrate the benefit of (fixed) slot registers and the contrastive loss. The paper discuss in detail how these components contribute to the performance gain and provides an additional mathematical justification for the contrastive loss.

**Strengths:**

* (S1) The paper is very well written, clearly explains the shortcomings of existing methods and motivates the proposed solutions. The methodology section is well-explained.
* (S2) Though the ideas in the paper can be seen as a combination of several ideas from existing works (register tokens [1], negative guidance [2], contrastive learning of slots [3]), the paper creatively combine these ideas in the context of slot attention methods. This enhances the technical contributions of the paper.
* (S3) The results for different tasks, from property prediction, object discovery, and compositional generation, clearly show that the method outperforms previous methods by a large margin. The improvement for categorical prediction is particularly impressive. Additionally, the paper provides detailed ablations for several configuration settings, including the number of register tokens and the scale of the CFG guidance weight.
* (S4) The results in Figure 7 are impressive, indicating that the slots actually bind to the objects and that the model is able to reconstruct the objects to a surprising degree with these slots. This highlights that the approach indeed allows to better disentangle the slot representations.

References:
* [1] Darcet et al., Vision Transformers Need Registers, ICLR 2024.
* [2] Karras et al., Guiding a Diffusion Model with a Bad Version of Itself, NeurIPS 2024.
* [3] Manasayan et al., Temporally Consistent Object-Centric Learning by Contrasting Slots, CVPR 2025.

**Weaknesses:**

* (W1) Fairness of evaluation — A significant issue in the comparison with other methods is the use of DINOv2 and the 512x512 image size for the training method. Other methods, such as SPOT and DINOSAUR, use a DINOv1 model with an image size much smaller than 512x512. Thus, the proposed gains in the method cannot be attributed solely to the slot attention registers and constructive loss.
* (W2) Ablations on the VOC dataset — Though I understand that performing the ablations on VOC is quicker, the VOC dataset is much simpler compared to the COCO dataset (many images consist of only a single object), and to thoroughly gain insights on the effect of the proposed changes, the paper should conduct the ablations on a more complex dataset like COCO (even if the ablations are done over a subset of the COCO dataset).
* (W3) The need for training of cross-attention (CA) layers is not clearly explained by Tab. 5. FG-ARI as a metric has been criticized by several papers (SPOT, GLASS), etc. The primary motivation for training the CA layers appears to be achieving a good reconstruction output rather than improving object discovery metrics. The paper should refer to Table 9 in the appendix to clarify the contribution of training CA layers.

Minor points:
1. Figure 2, lower branch should be $\epsilon_\bar{\theta}$ instead of $\epsilon_\{\theta}$.
2. Figure 2: DINO should be DINOv2.
3. Figure 6 is hard to understand due to the use of different input images with different layers. The figure should be supported with a more illustrative caption.
4. It would be useful if the paper were to provide more qualitative results for compositional generation.
5. In l. 236, the paper claims that $\mathbf{s}$ are DINOv2 features, but these are actually the slots computed from DINOv2 features.
6. Reference [2], see above, is missing.

**Questions:**

1. Can the authors discuss the effect of the method vs. the number of objects in the scene? For example, if a given image has a 7 objects for 7 slots, does adding register tokens hurt the performance in this case? What happens if the number of object tokens is greater than 7?
2. How did the authors settle on the ‘odd’ number of 77 register tokens? Where does this number come from? Also, does this large number of register tokens affect computational efficiency?
3. Can the authors discuss why the contrastive loss guidance scale needs to set to a really low value for good results?
4. For figure 7, it seems like the slots from StableLSD do not correspond/encode the object at all (we are not able to reconstruct the object from individual slot), however, the joint slot is able to reconstruct the scene fairly well. Why do you think this is case?
5. Is it really the case that fine-tuning the cross-attention mechanism (CA) does not reduce the expressive power of the model (l. 228)? How can we assure that this is indeed the case?

---

> ### Author Response · Authors · 2025-11-21
> **Response to Reviewer hjxk (1/3)**
>
> Thank you for your feedback and positive comments. They are very encouraging. Please find our responses below.
>
> ### [W1] Fairness of evaluation
> > Other methods, such as SPOT and DINOSAUR, use a DINOv1 model with an image size much smaller than 512x512. Thus, the proposed gains in the method cannot be attributed solely to the slot attention registers and constructive loss.
>
> Thank you for this remark. We agree that using a stronger backbone (DINOv2) and a higher training resolution can contribute to performance gains over methods such as SPOT and DINOSAUR. Our main claims, however, are also supported by ablations, all conducted under the same encoder and image-resolution settings, which isolate the effect of CODA’s architectural and training losses. Please also refer to our general comment on `Clarification on the experimental setup` for additional results. In addition, we downsample CODA’s outputs to 256×256 and recompute all reported FID/KID scores at this resolution, matching the setup used in prior works.
>
> ### [W2] Ablations on the VOC dataset
> >  the paper should conduct the ablations on a more complex dataset like COCO
>
> Thank you for the suggestion. We agree that ablations on COCO would provide additional insights. In the revised version, we’ve extended our ablation studies to COCO and summarized the results in Table 9.  CODA with all components still achieves the best overall results.
> | Reg | CA | CO | FG-ARI ↑ | mBOi ↑ | mBOc ↑ | mIoUi ↑ | mIoUc ↑ |
> | --- | --- | --- | --- | --- | --- | --- | --- |
> |  |  |  | 20.99 | 29.77 | 37.21 | 32.16 | 41.25 |
> | ✓ |  |  | 23.64 | 31.14 | 39.07 | 32.64 | 41.91 |
> |  | ✓ |  | 36.99 | 33.82 | 38.08 | 35.04 | 41.24 |
> |  |  | ✓ | 25.24 | 30.12 | 38.77 | 32.83 | **42.99** |
> | ✓ | ✓ |  | 45.95 | 35.80 | 40.32 | 35.76 | 41.75 |
> | ✓ | ✓ | ✓ | **47.54** | **36.61** | **41.43** | **36.41** | 42.60 |
>
> ### [W3] Reference to Appendix
> > The paper should refer to Table 9 in the appendix to clarify the contribution of training CA layers.
>
> We thank the reviewer for the suggestion. In the revised version, we explicitly refer to Table 9 (Appendix), which provides a more complete ablation: enabling CA training not only improves reconstruction quality but also consistently boosts multiple object-centric metrics (beyond FG-ARI).

---

> ### Author Response · Authors · 2025-11-21
> **Response to Reviewer hjxk (2/3)**
>
> ### [Q1] Effectiveness of CODA vs. the number of objects
> > Can the authors discuss the effect of the method vs. the number of objects in the scene?
>
> Thank you for this thoughtful question. The register slots are not intended to “compete” with semantic slots for objects; instead, they act as a residual buffer that mainly absorbs background and ambiguous mass.
>
> **When the number of objects ≤ number of semantic slots**: (e.g., 7 objects, 7 slots) In this case, adding register slots does not reduce the number of semantic slots available for objects: we still have 7 semantic slots plus a few additional register slots. Empirically, we observe that in such scenes register slots mostly capture background or “leftover” regions, and semantic slots still specialize to individual objects. We do not see a systematic performance drop in these cases compared to the variant without registers.
>
> **When the number of objects > number of semantic slots:** This is a general limitation of all slot-based object-centric models: if there are more objects than semantic slots, some objects must be grouped into the same slot. CODA does not remove this limitation, but register slots help reduce entanglement by taking on clutter and non-object regions, so that semantic slots are less burdened by background and overlapping evidence. Moreover, the contrastive alignment is defined only for semantic slots, which encourages them to explain object-level content, while register slots are discouraged from encoding object-like structure.
>
> In the revised version, we’ve added this short discussion to make this behavior and limitation more explicit (see Sec. F).
>
> ### [Q2] Selection for number of register slots
> >  How did the authors settle on the ‘odd’ number of 77 register tokens? does this large number of register tokens affect computational efficiency?
>
> - The choice of 77 register tokens is primarily driven by the text encoder of diffusion models. Stable Diffusion 1.5 uses a CLIP text encoder with a maximum sequence length of 77 tokens, and its cross-attention blocks are implemented assuming this length. By setting the number of register tokens to 77, we can reuse the embeddings of padding tokens as register slots (see our footnote L. 215).
> - As explained in our Appendix (see Sec. E.2), the computational overhead is relatively small. This is because the number of spatial tokens in the U-Net (from the 2D feature maps) is orders of magnitude larger than 77, so adding these registers has only a minor impact on the cost. In our setup, it marginally increases GPU time by **only 0.02%** compared to the baseline without using any register slot.
>
> ### [Q3] Contrastive loss guidance scale is small
> > Can the authors discuss why the contrastive loss guidance scale needs to set to a really low value for good results?
>
> Thank you for this question. Empirically, we observe that increasing $\lambda_\mathrm{cl}$ beyond this range can degrade visual quality and object-centric performance. We hypothesize that this happens because the contrastive term operates on slot-level features and, when heavily weighted, over-emphasizes alignment at the expense of the diffusion prior, leading to overspecialized and less realistic samples. In contrast, a small $\lambda_\mathrm{cl}$ acts as a regularizer that improves alignment while keeping the diffusion objective dominant.
>
> In the revised version, we’ve added a short discussion to clarify this (see Sec. E.4).
>
> ### [Q4] Explanation the behavior of Stable-LSD
> > the slots from StableLSD do not correspond/encode the object at all (we are not able to reconstruct the object from individual slot),  however, the joint slot is able to reconstruct the scene fairly well. Why?
>
> Thank you for this insightful observation. In Stable-LSD, slots are **only trained jointly** to reconstruct the full image: the diffusion loss is defined on the entire scene with all slots present, so the model learns a distributed code over slots rather than per-slot generations. Because cross-attention is soft, object information can be spread across multiple slots; when we remove all but one slot at test time, we put the model in an **out-of-distribution setting** it was never optimized for, so single-slot generations fail to capture coherent objects even though the combination of all slots can reconstruct the scene well. This reflects **slot entanglement** in Stable-LSD, where individual slots encode mixtures or fragments of multiple objects.
>
> One of the goals of CODA is precisely to reduce this entanglement. By using register slots and a contrastive alignment objective, we bias semantic slots to focus more on specific regions/objects and offload residual background to register slots. While this does not make single-slot generations perfect, it leads to substantially more object-specific content in the per-slot samples.
>
> In the revised version, we’ve added this clarification (see Sec. D).

---

> ### Author Response · Authors · 2025-11-21
> **Response to Reviewer hjxk (3/3)**
>
> ### [Q5] Finetuning cross-attention
> > Is it really the case that fine-tuning the cross-attention mechanism (CA) does not reduce the expressive power of the model (l. 228)? How can we assure that this is indeed the case?
>
> Thank you for this question. Our original wording at l.228 meant we do not modify the architecture of the diffusion model: the cross-attention layers we fine-tune are exactly the same modules as in the original backbone, so in principle the model’s representational capacity is unchanged.
>
> We agree that, from an optimization perspective, updating only CA layers does constrain how the model can adapt compared to fine-tuning all parameters. To check this empirically, we’ve compared (i) fine-tuning all parameters (FT) and (ii) updating only the cross-attention layers (CA) on the VOC dataset.
>
> | Model | FG-ARI ↑ | mBOi ↑ | mBOc ↑ | mIoUi ↑ | mIoUc ↑ |
> | --- | --- | --- | --- | --- | --- |
> | CA | **32.23** | **55.38** | **61.32** | **50.77** | **56.30** |
> | FT | 20.66 | 43.73 | 48.69 | 42.24 | 46.99 |
>
> As shown in this table, the CA-only variant actually performs better. Our interpretation is that the U-Net backbone is already well trained, and restricting updates to a small, targeted subset of parameters (the CA layers that mediate slot–image interaction) allows the model to adapt more effectively to the object-centric learning task, whereas full fine-tuning can overwrite useful pretrained structure and slightly hurt performance.
>
> ### [Q6] Qualitative results for compositional generation.
> > It would be useful if the paper were to provide more qualitative results for compositional generation.
>
> Thank you for this suggestion. In Fig. 12, we’ve extended compositional edits on real scenes, including (i) removing objects, (ii) swapping objects between images, (iii) adding objects, and (iv) changing backgrounds.
>
> ### [Q7] References
> > Reference [2], see above, is missing.
>
> Thank you for this suggestion. In the revised version, we make the connection to [2] more explicit in the related work (Section 2).

---

### Author Response · Authors · 2025-11-21
**Highlighted changes in the revised manuscript**

Thank you for your thorough reviews and insightful feedback. We are glad that the reviewers found our paper well motivated and well written (`hjxk`, `vEFV`).  They recognized that our approach is conceptually simple (`73eg`), but has strong/convincing/solid results (`hjxk`, `vEFV`, `73eg`) with comprehensive evaluations across multiple datasets. They also acknowledged the novelty and technical significance of our contributions (`hjxk`, `vEFV`). We also appreciate that reviewer `vEFV` found our analysis of mutual information interesting and inspiring.


We have carefully revised our submission to address your comments and suggestions. The updated manuscript has included the following changes (highlighted in blue).


**Main paper:**
- **Sec. 2:** Added missing references and expanded the related work as suggested. (`hjxk`, `73eg`)
- **Sec. 4.1:** Clarified the relationship to SlotAdapt and discussed the differences more explicitly. (`73eg`)
- **Sec. 5.3:** Updated FID/KID scores evaluated at 256×256 resolution. (`73eg`)
- **Sec. 5.4:** Added reference to an ablation study on COCO in Table 9. (`hjxk`)
- Corrected minor typos and improved wording. (`hjxk`, `73eg`)


**Appendix:**
- **Sec. B.4:** Clarified the experimental setup, including random seed variations and the image resolution used for FID/KID. (`73eg`)
- **Sec. D:** Added a discussion of residual background leakage in Fig. 7 and behaviour of Stable-LSD. (`vEFV`, `hjxk`)
- **Sec. E.3:** Included additional ablation studies on COCO for both segmentation and generation. (`hjxk`)
- **Sec. E.4:** Added a discussion explaining the choice of a small value for $\lambda_\mathrm{cl}$. (`hjxk`)
- **Sec. E.6:** Added comparisons with weakly supervised baselines (e.g., GLASS). (`vEFV`)
- **Sec. E.7:** Updated Fig. 9 with SlotAdapt visualizations and added further qualitative results on compositional editing, e.g., adding, removing, replacing objects, and changing backgrounds. (`hjxk`, `73eg`, `vEFV`)
- **Sec. F:** Added a discussion of CODA’s limitations (e.g., fixed number of slots, imperfect reconstruction) and outlined directions for future work, including extensions to other diffusion backbones. (`hjxk`, `73eg`, `vEFV`)

We appreciate your constructive feedback, which has significantly helped us strengthen and refine our manuscript.

---

> ### Author Response · Authors · 2025-11-21
> **Clarification on the experimental setup**
>
> Our choice of a higher resolution is primarily driven by the pretrained diffusion backbone: Stable Diffusion 1.5 was designed to operate optimally at 512×512 resolution  (see supplementary material for your reference). Since CODA only finetunes the cross-attention layers while keeping the rest frozen, generating at substantially different resolutions can significantly reduce the capacity of the pretrained model and degrade performance. Unlike CODA, other slot-based diffusion methods can alleviate this issue by introducing additional adapter layers or training from scratch for the diffusion backbone.
>
> To ensure a fair comparison in terms of image fidelity, we downsample CODA’s outputs to 256×256 and recompute all reported FID/KID scores at this resolution, matching the setup used in prior works. Since all methods use the same data and no additional supervision, we believe that our experiments constitute a meaningful and fair comparison.
>
> We attach an additional one-page PDF (`CODA_supplementary.pdf`) in the supplementary material with CODA’s results when trained at 256×256 resolution.

---

> ### Author Response · Authors · 2025-11-25
> **Additional experiments with more training budget**
>
> We thank the reviewers for their constructive feedback and helpful suggestions.
>
> To clarify the concern regarding training budget, we’ve conducted SlotAdapt under two additional training settings. Since official SlotAdapt checkpoints are not available, all reported numbers are obtained by running the authors’ code ourselves.
>
> - **With a substantially larger number of iterations (up to 500k steps)**. As shown in the tables below, more training iterations leads to only minor improvements and essentially saturates around the performance originally reported in their paper, **without closing the gap to CODA.**
>
>     *Segmentation results on VOC*
>     | Model | Resolution | # of iterations | FG-ARI ↑| mBOi ↑| mBOc ↑|
>     | --- | --- | --- | --- | --- | --- |
>     | CODA | 512x512 | 250k | **32.23** | **55.38** | **61.32** |
>     | SlotAdapt | 256x256 | 250k | 28.50 | 50.31 | 50.80 |
>     | SlotAdapt | 256x256 | 500k | 29.63 | 51.45 | 51.67 |
>     | *SlotAdapt (reported)* | 256x256 | 190k | 29.6 | 51.5 | 51.9 |
>
>     *Segmentation result on COCO*
>     | Model | Resolution | # of iterations | FG-ARI↑ | mBOi↑ | mBOc↑ |
>     | --- | --- | --- | --- | --- | --- |
>     | CODA | 512x512 | 500k | **47.54** | **36.61** | **41.43** |
>     | SlotAdapt | 256x256 | 250k | 43.64 | 33.20 | 37.21 |
>     | SlotAdapt | 256x256 | 500k | 42.30 | 34.03 | 38.31 |
>     | *SlotAdapt (reported)* | 256x256 | 250k | 41.4 | 35.1 | 39.2 |
>
>
> - **With higher resolution and the same training budget**. The table below shows that SlotAdapt does not benefit substantially from the higher resolution: on VOC, its performance at 512×512 is slightly worse than at 256×256, and on COCO it improves only in FG-ARI. **Even with this higher resolution, SlotAdapt still does not close the performance gap to CODA**.
>
>     *Segmentation results on VOC*
>
>     | Model | Resolution | # of iterations | FG-ARI ↑| mBOi ↑| mBOc ↑|
>     | --- | --- | --- | --- | --- | --- |
>     | CODA | 512x512 | 250k | **32.23** | **55.38** | **61.32** |
>     | SlotAdapt | 512x512 | 250k | 27.83 | 50.83 | 50.66 |
>     | *SlotAdapt (reported)* | 256x256 | 190k | 29.6 | 51.5 | 51.9 |
>
>     *Segmentation result on COCO*
>     | Model | Resolution | # of iterations | FG-ARI ↑| mBOi ↑| mBOc ↑|
>     | --- | --- | --- | --- | --- | --- |
>     | CODA | 512x512 | 500k | **47.54** | **36.61** | **41.43** |
>     | SlotAdapt | 512x512 | 500k | 44.52 | 34.25 | 38.36 |
>     | *SlotAdapt (reported)* | 256x256 | 250k | 41.4 | 35.1 | 39.2 |
>
>
> These results further support that **CODA’s gains stem from fundamentally different design choices**, rather than from engineering refinements such as more training iterations or higher image resolution. Even with a larger number of trainable parameters (~3x), SlotAdapt still underperforms CODA.
>
> | Model | # of parameters |
> | --- | --- |
> | SlotAdapt | 116.83 M |
> | CODA | 39.85 M |

---

### Author Response · Authors · 2025-11-28
**Thank you**

Dear reviewers and area chairs,

Thank you again for the detailed reviews and for acknowledging the strengthened experiments and appendix. We agree that CODA is inspired by prior work, and we do not claim to have invented registers or contrastive loss. Our main goal is to **explicitly address two persistent issues in existing slot diffusion models**—slot entanglement and weak slot–image alignment—which we demonstrate remain significant even for SlotAdapt (e.g., incoherent single-slot generations). CODA tackles these issues through a **coupled design**: (i) context-free, strictly non-semantic register slots that are used as attention sinks, and (ii) a contrastive alignment loss applied only to semantic slots. This combination yields substantially cleaner single-slot generations, stronger property prediction, and more robust compositional behavior under matched backbones and extended training. For a more detailed discussion, please refer to "**Response to Reviewer 73eg (1/4)**" of "**Regarding contributions**".

While CODA is not entirely orthogonal to prior work, we see these design and analysis choices as a focused, well-justified step within this research line, and we hope they clarify the empirical improvements we observe. We also hope this clarification is helpful for your assessment of the manuscript.

Please let us know if any further clarification or results would be needed. Thank you again for the detailed feedback and the time spent helping us improve the manuscript.

---

### Author Response · Authors · 2025-12-03
**Rebuttal summary to ACs**

Dear Area Chairs,

to facilitate your assessment, we provide below a brief summary of our responses.

## Reviewer hjxk

**Concerns:** Evaluation fairness, limited ablations (mostly VOC), and more qualitative results.

**Actions taken:**
- **Evaluation fairness:** Clarified the evaluation protocol that CODA is trained at 512×512 and downsampled to 256×256 for FID/KID to match prior work; all methods use the same data and no additional supervision.
- **Resolution clarification:** Ran CODA at 256x256 showing it remains better on FG-ARI at 256×256, while the main results continue to use 512×512 generation with 256×256 evaluation to avoid SD-1.5’s resolution mismatch and keep the comparison fair.
- **Limited ablations:** Added ablation studies on the COCO dataset (App. E3, Tab. 9) to complement the VOC ablations.
- **Qualitative and limitation discussion:** Included more qualitative compositional edits (Fig. 12), an explanation of Stable-LSD’s behavior (App. D), and a discussion of limitations when the number of objects and slots do not match (App. F).

## Reviewer vEFV

**Concerns:** Comparison with GLASS, CODA with a larger backbone (SD.2.1), visual reconstruction comparison, and more training iterations for SlotAdapt.

**Actions taken:**
- **Comparison with GLASS:** Added a comparison with GLASS and a GLASS† variant using class labels (App. E.6, Tab. 12).
- **CODA with a larger backbone:** Reported results for CODA with SD.2.1 as backbone.
- **Reconstruction comparison:** Included a direct visual comparison of reconstruction quality between SlotAdapt and CODA (App. E.7, Fig. 10).
- **More training iterations:** Trained SlotAdapt for substantially longer (up to 500k steps); its performance saturates and still does not approach CODA, addressing the concern that CODA’s gains might come purely from more training.
- **Additional discussion:** Added a discussion of residual background leakage in the qualitative results (App. D).

## Reviewer 73eg

**Concerns:** Evaluation protocol, how different CODA is from SlotAdapt and Learning-to-Compose, more qualitative results

**Actions taken:**
- **Evaluation protocol:** Clarified that all results are averaged over five random seeds; all reported FID/KID use 512×512 outputs downsampled to 256×256; and all methods are trained and evaluated on the same datasets with no additional supervision.
- **Training fairness:** Trained SlotAdapt at 512×512 under the same backbone family as CODA; this yields a small FG-ARI improvement but SlotAdapt still clearly underperforms CODA, addressing the concern that CODA's gains might come purely from using 512×512.
- **Relation to prior work:** Added explicit related-work discussion for SlotAdapt (Sec. 4.1) and Learning-to-Compose (Sec. 2), clarifying that
SlotAdapt’s register token is context-dependent (global context from SA), whereas CODA uses context-free, non-semantic register slots that are never used for prediction or alignment.
CODA’s contrastive loss is an alignment objective, not a likelihood over random slot mixtures as in Learning-to-Compose.
- **Qualitative and limitations:** Added more qualitative compositional edits (Fig. 12) and a detailed limitations section (Sec. F) acknowledging that some edits remain fragile in complex scenes and that identity preservation is not yet perfect.

---

### Meta-Review · Area_Chair_WBkh · 2026-01-11

**Summary:**

Reviewers generally agreed that the paper is well written, well motivated, and supported by strong empirical results. The main concerns focused on
1. degree of novelty relative to prior slot-based diffusion methods, SlotAdapt, Learning-to-Compose
2. fairness of comparisons, particularly regarding image resolution, training budget, and backbone design

While some reviewers viewed the contributions as incremental, they acknowledged that the proposed design leads to clear and consistent improvements in object discovery, property prediction, and compositional generation on both synthetic and real-world datasets. Overall, these concerns do not outweigh the strength of the results, and they support acceptance.

**Reviewer Concerns:**

The rebuttal addressed the major substantive concerns in a thorough and convincing manner. The authors clarified the evaluation protocol by reporting 512×512 generation with 256×256 FID/KID, added ablations on the more challenging COCO dataset, and conducted extended training and higher-resolution experiments for SlotAdapt to show that its performance saturates. They also included comparisons with GLASS, additional experiments using SD 2.1, expanded qualitative results for compositional editing, and reported results averaged over multiple random seeds. These additions significantly strengthen the empirical case and make it difficult to attribute the gains solely to resolution, training budget, or backbone differences.

The remaining concerns are primarily about conceptual novelty, as some components build on existing ideas. However, these are largely questions of positioning rather than technical soundness. While the individual components relate to prior work, their combination is simple, effective, and well validated empirically.

**Reviewer Scores:**

i don't think there'd be much change

---

### Decision · Program_Chairs · 2026-01-26

Accept (Poster)